# SECOND-ORDER BOUNDS FOR [0,1]-VALUED REGRESSION VIA BETTING LOSS

## ABSTRACT

We consider the $[0, 1]$-valued regression problem in the stochastic setting. In a related problem called cost-sensitive classification, Foster and Krishnamurthy (2021) have shown that the log loss minimizer achieves an improved generalization bound compared to that of the squared loss minimizer in the sense that the bound scales with the cost of the best classifier, which can be arbitrarily small depending on the problem instance. Such a result is often called a first-order bound. For $[0, 1]$-valued regression, we first show that the log loss minimizer leads to a similar first-order bound. We then ask if there exists a loss function that achieves a variance-dependent bound (also known as a second-order bound), which would be a strict improvement upon first-order bounds. We answer this question in the affirmative by proposing a novel loss function called betting loss. Our result is *variance-adaptive* in the sense that the bound is attained by an algorithm *without any knowledge about the variance*, which is in contrast to weighted least squares with known variances or those that model label variance or its distribution such as distributional reinforcement learning.

## 1 INTRODUCTION

We consider the $[0, 1]$-valued regression problem: We are given a dataset $D_n = \{(x_t, y_t)\}_{t=1}^n$ where $x_t \in \mathcal{X}$ is the feature of the $t$-th data point and $y_t \in [0, 1]$ is its label. We assume the data $(x_t, y_t) \sim \mathcal{D}_{X,Y}$ is i.i.d., $\forall t \in [n]$. The goal is to, given a function class $\mathcal{F} \subset \{\mathcal{X} \to [0, 1]\}$, find a function $f$ such that the prediction $f(x)$ is as close as possible to $y$ on average where $(x, y) \sim \mathcal{D}_{X,Y}$.

While being one of the simplest machine learning tasks, this regression task applies to numerous practical applications. First, classification is a special case of this problem where the label space is $\mathcal{Y} = \{0, 1\}$. Second, in Reinforcement Learning (RL), the rewards are typically bounded, and when the episode length is upperbounded, the cumulative reward per episode is also bounded. Thus, in the function approximation setting, one can easily scale the cumulative rewards from each state-action to $[0, 1]$ and perform regression. With this regression, one can construct a policy that choose the action with the highest predicted value. In goal-oriented RL, regardless of the length of the episode, the rewards are given only at the end of the episode, so, as long as the reward is bounded in a fixed interval, $[0, 1]$-valued regression applies. Finally, human preferences can mostly be expressed as a value in $[0, 1]$. For example, 5-star ratings ($\in \{1, 2, 3, 4, 5\}$) for products can be affine-transformed to $[0, 1]$. Furthermore, datasets commonly used for aligning Large Language Models (LLMs) such as HelpSteer2 originally contain scores $\{0, 1, 2, 3, 4\}$ (Wang et al., 2024b). Therefore, despite being simple and rather elementary, $[0, 1]$-valued regression is still important, and theoretical and algorithmic advancements can potentially have a huge impact in practice.

What do we know about the fundamental performance limits of $[0, 1]$-valued regression? The de facto standard regression algorithm is to simply minimize the squared loss. However, it is not clear at all if squared loss is optimal for $[0, 1]$-valued regression. In a related problem called cost-sensitive classification, Foster and Krishnamurthy (2021) have shown that the squared loss is not optimal for $[0, 1]$-valued costs. Instead, they have shown that the log loss achieves a strictly improved performance bound, a rate that is provably not attained by the squared loss (Foster and Krishnamurthy, 2021, Theorem 2). Specifically, their bound is of the *first-order* type, which means that the performance bound scales with the *magnitude* of the cost/reward being accumulated by the optimal policy. Such a bound is never worse than the standard worst-case bound, yet can be much smaller depending on

the problem instance. This has also been called small-loss bound and can be viewed as a problem-dependent accelerated rate.

Such a first-order bound appeared in various machine learning problems (Freund and Schapire, 1997; Foster and Krishnamurthy, 2021; Wagenmaker et al., 2022). In these problems, there is another concept called *second-order* bound (Cesa-Bianchi et al., 2007). While the precise definition can vary across problems, when making stochastic assumptions about how $y$ is related to $x$, it means that the bound scales with the label's second moment or the variance, which can be much smaller than the magnitude of the label. We elaborate more on this in Section 5.

Motivated by the fact that Foster and Krishnamurthy (2021) simply switched a loss function to obtain a first-order bound in cost-sensitive classification, we first report that the same is true in $[0, 1]$-valued regression (see Theorem 1 in Section 2). Given this positive answer, we take a step further and ask the following research question:

*Does there exist a loss function whose minimizer leads to a second-order bound?*

In this paper, we provide an affirmative answer by proposing a novel loss function inspired by the betting-based confidence set (Waudby-Smith and Ramdas, 2023; Orabona and Jun, 2024). We emphasize that our algorithm does not require conditional variances as input and allows them to be arbitrarily different. This is in stark contrast to some existing work that either requires the variance as input (Zhao et al., 2023b) or models variance as part of function approximation (Wang et al., 2024a; Weltz et al., 2023). In some sense, our result shows that obtaining second-order bounds (i.e., adapting to unknown variances) is a free lunch, statistically speaking, in the sense that we do not have to model variance to adapt to it. While there are works that achieve second-order bounds without knowledge of the conditional variances (Zhao et al., 2024; Jun and Kim, 2024; Jia et al., 2024; Pacchiano, 2025), the tools therein are specialized for their own contextual bandit problem and do not naturally imply an estimator for $[0, 1]$-valued regression. We discuss more related work in Section 5. Moreover, in Section 4, our experimental results demonstrate that the proposed loss consistently achieves lower mean absolute error (MAE) than both the log loss and the squared loss.

## 2 PRELIMINARIES

**Notations.** We denote $f_x := f(x)$ for any function $f$ and any $x \in \mathcal{X}$. We adopt the nonasymptotic version of $\lesssim$; i.e., $f(x) \lesssim g(x)$ means that there exists an numerical constant $c > 0$, s.t. $f(x) \leq c \cdot g(x), \ \forall x$.

**Regression with $[0, 1]$-valued label.** We consider the standard supervised learning setting with bounded regression targets. Let $\mathcal{X}$ denote the input space. We observe a dataset $D_n = \left\{ (x_t, y_t) \right\}_{t=1}^n$ where each pair $(x_t, y_t)$ is drawn i.i.d. from an unknown distribution $\mathcal{D}_{X,Y}$ over $\mathcal{X} \times [0, 1]$. We denote by $\mathcal{D}_{Y|X}$ the distribution of the label conditioning on the input, and $\mathcal{D}_X$ the marginal distribution of the input.

Let $\mathcal{F} \subset \left\{ \mathcal{X} \to [0, 1] \right\}$ be a class of prediction functions mapping inputs to the unit interval. We assume *realizability*, i.e., there exists a function $f^* \in \mathcal{F}$ such that:
$$\mathbb{E}_{y \sim \mathcal{D}_{Y|X}}[y \mid x] = f^*(x), \text{ for all } x \in \mathcal{X}.$$

Note that, since we do not have further restrictions on $\mathcal{D}_{X,Y}$, the conditional variance $\sigma_x^2 := \mathbb{E}_{y \sim \mathcal{D}_{Y|X}}[(y - f^*(x))^2 \mid x]$ can vary across $x \in \mathcal{X}$.

Given the observed data $D_n$, the learning goal is to find a hypothesis $\hat{f} \in \mathcal{F}$ that achieves low expected absolute error with respect to the ground-truth regression function $f^*$:
$$\mathbb{E}_{x \sim \mathcal{D}_X} \left[ |f^*(x) - \hat{f}(x)| \right].$$

The goal is to bound such a generalization error of the learned hypothesis in terms of the sample size $n$, the function class complexity $\ln |\mathcal{F}|$, and the confidence level $\delta$.

The de facto standard algorithm for regression is the squared loss minimizer:
$$\hat{f} = \arg\min_{f \in \mathcal{F}} \ \frac{1}{n} \sum_{(x,y) \in D_n} \frac{1}{2}(f(x) - y)^2 \tag{1}$$

A classical result on the squared loss minimizer yields the following:

$$\mathbb{E}_{x \sim \mathcal{D}_X} \left[ |f^*(x) - \hat{f}(x)| \right] \lesssim \sqrt{\frac{\ln(|\mathcal{F}|/\delta)}{n}}.$$

While this bound is simple and general, it does not incorporate any notion of conditional variance. It treats all inputs as equally noisy, making it inherently variance-insensitive and potentially loose in heterogeneous noise settings.

A recent result on the log loss minimizer (Foster and Krishnamurthy, 2021, Theorem 3) immediately implies the following first-order generalization bound, which scales with the magnitude of the target regression function $f^*(x)$ and its complement $1 - f^*(x)$ in expectation:

$$\mathbb{E}_{x \sim \mathcal{D}_X} \left[ |f^*(x) - \hat{f}(x)| \right] \lesssim \sqrt{\frac{\left( \mathbb{E}_x[f^*(x)] \wedge \mathbb{E}_x[1 - f^*(x)] \right) \cdot \ln(|\mathcal{F}|/\delta)}{n}} + \frac{\ln(|\mathcal{F}|/\delta)}{n}. \quad (2)$$

In the following theorem, we further improve the bound above to scale with $\mathbb{E}_x[f^*(x)(1 - f^*(x))]$.

**Theorem 1.** *With probability at least $1 - \delta$,*

$$\mathbb{E}_{x \sim \mathcal{D}_X} \left[ |f^*(x) - \hat{f}(x)| \right] \lesssim \sqrt{\frac{\mathbb{E}_x[f^*(x)(1 - f^*(x))] \cdot \ln(|\mathcal{F}|/\delta)}{n}} + \frac{\ln(|\mathcal{F}|/\delta)}{n}.$$

We note that this bound strictly improves upon the immediate implication of Foster and Krishnamurthy (2021) (Eqn. (2)), as $\mathbb{E}_x[f^*(x) \wedge (1 - f^*(x))] \leq \mathbb{E}_x[f^*(x)] \wedge \mathbb{E}_x[1 - f^*(x)]$, the gap between these two quantities can be arbitrarily large as we show in Appendix F.

The bound in Theorem 1 depends on the variance proxy $f^*(x)(1 - f^*(x))$, which upper bounds the conditional variance $\sigma_x^2 = \mathbb{E}[(y - f^*(x))^2 \mid x]$, as indicated by the following Lemma.

**Lemma 2.** *Let $Y \in [0, 1]$ be a random variable. Then $\mathrm{Var}(Y) \leq \mathbb{E}[Y](1 - \mathbb{E}[Y])$, and the equality is attained iff $Y$ is Bernoulli-distributed.*

However, the variance proxy can be very loose, especially when $y \mid x$ is not Bernoulli. In many applications, such as those mentioned in Section 1, the label distributions are often heteroscedastic and non-Bernoulli: some inputs yield lower uncertainty on $Y$ conditioning on $X$ (i.e., low variance), while others are more uncertain (i.e., high variance), and the outputs can take values other than the boundary points 0 and 1. In such settings, first-order bounds may fail to capture the true learnability of the problem.

To address this, our objective in this work is to derive *second-order generalization bounds* that adapt to the true conditional variance. Specifically, we aim to obtain bounds of the form:

$$\mathbb{E}_{x \sim \mathcal{D}_X} \left[ |f^*(x) - \hat{f}(x)| \right] \lesssim \sqrt{\frac{\mathbb{E}_x[\sigma_x^2] \cdot \ln(|\mathcal{F}|/\delta)}{n}} + \frac{\ln(|\mathcal{F}|/\delta)}{n},$$

which provides tighter guarantees in settings where $\mathcal{D}_X$ places a nontrivial probability on $x$ such that the conditional variance $\sigma_x^2$ is much smaller than the worst-case upper bound $f^*(x)(1 - f^*(x))$, without requiring the variances as input to the algorithm.

## 3 SECOND-ORDER BOUND VIA BETTING LOSS

We propose a novel regression algorithm that adapts to conditional variance by minimizing a worst-case form of betting loss, a loss function originally motivated by coin-betting frameworks. The goal is to learn a hypothesis $\hat{f}$ from a function class $\mathcal{F}$, using a dataset $D_n = \{(x_t, y_t)\}_{t=1}^n$ of feature-label pairs, such that $\hat{f}$ achieves strong generalization guarantees that adapt to the heteroskedastic noise structure of the problem.

At the core of our algorithm is a robust min-max optimization that seeks a function $f \in \mathcal{F}$ whose performance is stable against perturbations in the direction of any other hypothesis $h \in \mathcal{F}$. To this end, we define:

- A fixed parameter $\overline{\phi} := \frac{n}{4}$ that controls the magnitude of perturbation.
- A clipped betting loss function:

$$H_{\phi,c}(h, f) := \sum_{(x,y) \in D_n} \ln\left(1 + (y - f_x)\overline{(\phi(h_x - f_x))}_{[-c,c]}\right)$$

where $\overline{(x)}_{[a,b]} := \max\{\min\{x, b\}, a\}$ and $c \in [0, \frac{1}{4}]$ is a clipping threshold.

This formulation ensures that $\hat{f}$ performs well even under worst-case (clipped) perturbations in the direction of any other hypothesis $h \in \mathcal{F}$, across all allowed magnitudes $\phi$ and clipping levels $c$. Algorithm 1 summarizes the procedure.

---

**Algorithm 1** Variance-Adaptive Regression via Betting Loss Minimization

---

**Require:** Dataset $D_n = \{(x_t, y_t)\}_{t=1}^n$, hypothesis class $\mathcal{F}$
  1: Compute

$$\hat{f} := \arg\min_{f \in \mathcal{F}} \max_{h \in \mathcal{F}} \max_{\phi \in [0, \overline{\phi}]} \max_{c \in [0, \frac{1}{4}]} \frac{1}{n} H_{\phi,c}(h, f) .$$

  2: **return** $\hat{f}$

---

The objective minimized by Algorithm 1 – the worst-case clipped betting loss – directly governs the generalization behavior of the learned hypothesis. To formalize this, define the functional:

$$L_n(f) := \max_{h \in \mathcal{F}} \max_{\phi \in [0, \overline{\phi}]} \max_{c \in [0, \frac{1}{4}]} \frac{1}{n} \sum_{(x,y) \in D_n} \ln\left(1 + (y - f_x)\overline{(\phi(h_x - f_x))}_{[-c,c]}\right)$$

which exactly matches the objective minimized by the algorithm. The following theorem provides a high-probability bound on the expected absolute error of any $f \in \mathcal{F}$ in terms of this loss.

**Theorem 3** (Finite class). *There exists numerical constants $c_1, c_2$ and $c_3$, such that with probability at least $1 - \delta$,*

$$\forall f \in \mathcal{F}, \ \mathbb{E}_x |f_x - f_x^*| \leq c_1 \cdot \sqrt{\mathbb{E}\,\sigma_x^2 \cdot \left(\frac{1}{n} \ln\left(\frac{|\mathcal{F}|n}{\delta}\right) + \max\{L_n(f) - L_n(f^*), 0\}\right)}$$

$$+ c_2 \cdot \frac{1}{n} \ln\left(\frac{|\mathcal{F}|n}{\delta}\right) + c_3 \cdot (L_n(f) - L_n(f^*))$$

We provide the full proof in Appendix B.

This result provides a high-probability bound on the prediction error $\mathbb{E}_x |f_x - f_x^*|$ of any $f \in \mathcal{F}$, in terms of the difference in empirical betting loss $L_n(f) - L_n(f^*)$. Crucially, the bound adapts to the conditional variance $\sigma_x^2$ in the leading term. The excess betting loss $L_n(f) - L_n(f^*)$ directly controls the mean absolute error. In particular, applying the theorem to the output $\hat{f} = \arg\min_{f \in \mathcal{F}} L_n(f)$ of Algorithm 1, we obtain:

$$\mathbb{E}_x |\hat{f}_x - f_x^*| \leq c_1 \cdot \sqrt{\mathbb{E}\,\sigma_x^2 \cdot \left(\frac{1}{n} \ln\left(\frac{|\mathcal{F}|n}{\delta}\right)\right)} + c_2 \cdot \frac{1}{n} \ln\left(\frac{|\mathcal{F}|n}{\delta}\right)$$

This bound reflects a variance-adaptive fast rate, which improves over convergence bounds that scale with the worst-case noise $f_x^*(1 - f_x^*)$. In particular, when the conditional variance $\sigma_x^2$ is small on average, the mean absolute error of $\hat{f}$ becomes correspondingly small – without requiring prior knowledge of the noise structure. This establishes Algorithm 1 as a variance-adaptive learning procedure for [0,1]-valued regression.

Theorem 3 provides a general excess risk bound that holds uniformly for all $f \in \mathcal{F}$ over any finite hypothesis class $\mathcal{F}$. Moreover, by combining this result with complexity control via covering numbers,

we can derive concrete generalization bounds for a broader family of hypothesis classes characterized by polynomial covering numbers. Following common terminology (see, e.g., Rakhlin et al. (2017)), we will refer to such VC-type classes as parametric classes.

**Definition 4** (Parametric class). *A class of functions $\mathcal{F}$ is a parametric class if there exists positive constants $A$ and $v$, such that for every $\varepsilon > 0$, the covering number $N(\varepsilon, \mathcal{F}, \|\cdot\|_\infty)$ satisfies the inequality:*

$$N(\varepsilon, \mathcal{F}, \|\cdot\|_\infty) \leq (\frac{A}{\varepsilon})^v . \tag{3}$$

**Theorem 5** (Parametric class). *Assume the covering number of $\mathcal{F}$ satisfies Eqn. (3). Then, there exists constants $c_1$ and $c_2$, such that with probability at least $1 - \delta$, the output $\hat{f} = \arg\min_{f \in \mathcal{F}} L_n(f)$ satisfies:*

$$\mathbb{E}_x |\hat{f}_x - f_x^*| \leq c_1 \cdot \sqrt{\mathbb{E}_x \sigma_x^2 \frac{v}{n} \ln(\frac{n}{\delta})} + c_2 \cdot \frac{v}{n} \ln(\frac{n}{\delta}) .$$

This result follows from Theorem 3 by applying polynomial covering numbers of parametric classes. We present the full proof in the appendix C.

As our work establishes generalization results for parametric classes, it is desirable to ground our theory on standard example function classes. We demonstrate that the standard linear function class fits this definition, confirming that our results have direct implications for standard ML tasks like linear regression.

**Corollary 6** (Linear class). *Let $\mathcal{F}$ be a linear function class in $d$-dimensional space: $\mathcal{F} = \{x \mapsto x^\top \theta + \frac{1}{2} : \|\theta\|_2 \leq \frac{1}{2}\}$ and $\mathcal{X}$ be the instance space: $\mathcal{X} = \{x \in \mathbb{R}^d : \|x\|_2 \leq 1\}$. Then, there exist constants $c_1$ and $c_2$, such that with probability at least $1 - \delta$, the output $\hat{f} = \arg\min_{f \in \mathcal{F}} L_n(f)$ satisfies:*

$$\mathbb{E}_x |\hat{f}_x - f_x^*| \leq c_1 \cdot \sqrt{\mathbb{E}_x \sigma_x^2 \frac{d}{n} \ln(\frac{n}{\delta})} + c_2 \cdot \frac{d}{n} \ln(\frac{n}{\delta}).$$

Ignoring $\mathbb{E}_x \sigma_x^2$, the dominant term in our bound attains the minimax-optimal rate for linear regression in $d$-dimensional spaces. This aligns with classical results showing that, even in the well-specified setting with Gaussian noise, no estimator can achieve a faster worst-case rate than $\mathcal{O}(\sqrt{\frac{d}{n}})$ for the prediction error of $\mathbb{E}_x |\hat{f}_x - f_x^*|$ (Tsybakov, 2004; Wainwright, 2019), matching our bound up to logarithmic factors.

This corollary illustrates the generality of the proposed framework. Notably, our bound is adaptive to the conditional variance of the label while matching the worst-case guarantees of classical approaches for the linear class.

**Does it work for nonparametric classes?** It is natural to ask whether our guarantees extend to nonparametric function classes. However, all our attempts based on standard techniques did not lead to a valid bound. The key difficulty seems to be that the loss function $L_n$ can be $O(n)$-Lipschitz in the input $f$, unlike squared loss, which is $O(1)$-Lipschitz in the input $f$. We conjecture that our betting loss minimizer is not consistent for nonparametric classes in general. We provide a related observation that may support our conjecture in the Appendix E.

## 4 EXPERIMENTS

In this section, we provide the empirical results that confirm our theoretical findings.

**Function class.** We consider a $d$-dimensional regression setting with target function $f^*$. First, we sample $\theta^* \in \mathbb{R}^d$ from an isotropic Gaussian distribution and normalize it to have Euclidean norm $S = 0.5$:

$$\theta^* \sim \mathcal{N}(0, I_d), \qquad \theta^* \leftarrow \frac{S}{\|\theta^*\|} \text{ followed by } \theta^*.$$

The target function is then defined as

$$f^* := (x \mapsto \sigma(x^T \theta^*)),$$

where $\sigma(z) := 1/(1 + \exp(-z))$ is the sigmoid function.

To avoid the potential complications from the convergence issues from optimization, we consider a finite function class. Specifically, we sample 20 $\theta$'s independently by drawing $\eta$ from the unit sphere followed by setting

$$\theta = \theta^* + \varepsilon \cdot \eta$$

where $\varepsilon = 0.2$. Let $\Theta$ be the set of these $\theta$'s and $\theta^*$, which means $|\Theta| = 21$. Finally, we construct our function class as

$$\mathcal{F} := \{x \mapsto \sigma(x^T \theta) : \theta \in \Theta\}.$$

**Data distribution.** Feature vectors are sampled as

$$x \sim \frac{1}{\sqrt{d}} \mathcal{N}(0, I_d).$$

For each $x$, we generate its label from a Beta distribution with mean $f_x^*$ and parameter $\rho \in (0, 1)$:

$$y \sim \text{Beta}\left(f_x^* \cdot \frac{1-\rho}{\rho}, \ (1 - f_x^*) \cdot \frac{1-\rho}{\rho}\right),$$

which satisfies that $y \in [0, 1]$ with probability 1.

Let

$$\alpha = f_x^* \cdot \frac{1-\rho}{\rho}, \qquad \beta = (1 - f_x^*) \cdot \frac{1-\rho}{\rho}.$$

For a $\text{Beta}(\alpha, \beta)$ distribution, the variance is

$$\text{Var}(y) = \frac{\alpha\beta}{(\alpha+\beta)^2(\alpha+\beta+1)}.$$

Plugging in $\alpha = f_x^* \frac{1-\rho}{\rho}$ and $\beta = (1 - f_x^*)\frac{1-\rho}{\rho}$ gives

$$\alpha\beta = f_x^*(1 - f_x^*)\left(\frac{1-\rho}{\rho}\right)^2.$$

Also,

$$(\alpha+\beta)^2(\alpha+\beta+1) = \left(\frac{1-\rho}{\rho}\right)^2\left(\frac{1-\rho}{\rho} + 1\right).$$

Therefore,

$$\text{Var}(y) = \frac{f_x^*(1 - f_x^*)\left(\frac{1-\rho}{\rho}\right)^2}{\left(\frac{1-\rho}{\rho}\right)^2\left(\frac{1-\rho}{\rho} + 1\right)}.$$

By simplifying the expression, we obtain the following result:

$$\text{Var}(y) = f_x^*(1 - f_x^*)\rho.$$

Therefore, the variance is proportional to $\rho$.

**Training.** Given the candidate set $\mathcal{F}$, we select a function by minimizing either log loss, squared loss, or betting loss on the training data $D_n \sim \mathcal{D}_{X,Y}^n$.

The log loss of $f$ is

$$L_n^{\log}(f) = -\frac{1}{n} \sum_{(x,y)\in D_n} \left(y \log f_x + (1 - y) \log(1 - f_x)\right),$$

and the selected function is

$$\hat{f}^{\log} = \arg\min_{f \in \mathcal{F}} L_n^{\log}(f).$$

Similarly, the squared loss of $f$ is

$$L_n^{\text{squared}}(f) = \frac{1}{n} \sum_{(x,y) \in D_n} (y - f_x)^2$$

and the selected function is

$$\hat{f}^{\text{squared}} = \arg\min_{f \in \mathcal{F}} L_n^{\text{squared}}(f).$$

For betting loss, we define

$$L_n^{\text{betting}}(f) = \max_{h \in \mathcal{F}} \max_{\phi \in [0,\overline{\phi}]} \max_{c \in [0,1]} \frac{1}{n} \sum_{(x,y) \in D_n} \ln\Big(1 + (y - f_x)\,\overline{(\phi(h_x - f_x))}_{[-c,c]}\Big),$$

where $\overline{(z)}_{[a,b]} := \max\{\min\{z,b\},a\}$, and $\overline{\phi} = n/4$. In our implementation, we discretize $\phi \in \{1, 2, 4, \ldots, \frac{n}{4}\}$ and $c \in \{\frac{1}{n}, \frac{2}{n}, \frac{4}{n}, \ldots, 1\}$ to solve the inner maximization, and select

$$\hat{f}^{\text{betting}} = \arg\min_{f \in \mathcal{F}} L_n^{\text{betting}}(f).$$

**Evaluation.** We evaluate the performance of the trained function $\hat{f}$ using the mean absolute error (MAE):

$$\text{MAE}(\hat{f}) = \mathbb{E}_{(x,y) \sim \mathcal{D}_{X,Y}}[|\hat{f}(x) - y|]$$

We perform Monte Carlo estimate of the MLE using a test set of size $m = 10{,}000$.

We fix the feature dimension to $d = 2$, vary the training sample-to-dimension ratio $n/d \in \{2, 4, 8\}$, and try $\rho \in \{0.01, 0.02, 0.04\}$. For each configuration $(n/d, \rho)$, we repeat each experiment 200 times, and we report the average MAE and its standard error.

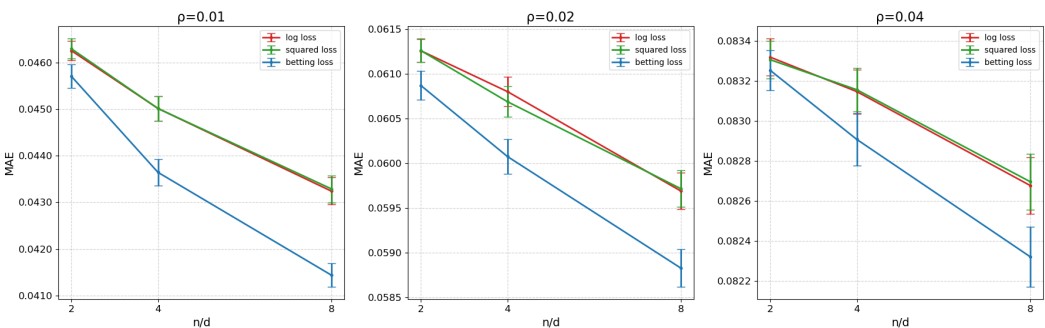

Figure 1: Comparison of average mean absolute error (MAE) obtained under log loss, squared loss and betting loss across varying $n/d$ and $\rho$. Error bars denote the standard error

**Results.** Table 1 and Figure 1 summarize the average MAE with standard error from function selection with log loss, squared loss and betting loss across different values of $n/d$ and $\rho$. The results show that all three losses decrease as $n/d$ increases, with betting loss consistently achieving the lowest MAE.

## 5 RELATED WORK

**Regression with heteroscedastic noise.** Regression with heteroscedastic noise can be dated back to (Aitkin, 1935), and has been further developed into Gaussian processes (Kersting et al., 2007; Goldberg et al., 1997). However, these works assume knowledge of the variances.

**First- and second-order bounds.** From the adversarial setting, to our knowledge, the first appearance of the first-order bound is from the prediction with expert advice setting Freund and Schapire (1997),

| $n/d$ | $\rho$ | Log loss MAE | Squared loss MAE | Betting loss MAE |
|---|---|---|---|---|
| | 0.01 | $0.04626 \pm 0.00021$ | $0.04630 \pm 0.00021$ | $0.04571 \pm 0.00025$ |
| 2 | 0.02 | $0.06126 \pm 0.00013$ | $0.06126 \pm 0.00013$ | $0.06087 \pm 0.00016$ |
| | 0.04 | $0.08332 \pm 0.00009$ | $0.08331 \pm 0.00009$ | $0.08325 \pm 0.00010$ |
| | 0.01 | $0.04502 \pm 0.00027$ | $0.04501 \pm 0.00027$ | $0.04364 \pm 0.00029$ |
| 4 | 0.02 | $0.06080 \pm 0.00017$ | $0.06069 \pm 0.00017$ | $0.06008 \pm 0.00020$ |
| | 0.04 | $0.08315 \pm 0.00011$ | $0.08315 \pm 0.00011$ | $0.08291 \pm 0.00013$ |
| | 0.01 | $0.04324 \pm 0.00029$ | $0.04329 \pm 0.00029$ | $0.04144 \pm 0.00025$ |
| 8 | 0.02 | $0.05969 \pm 0.00021$ | $0.05972 \pm 0.00020$ | $0.05883 \pm 0.00021$ |
| | 0.04 | $0.08268 \pm 0.00014$ | $0.08270 \pm 0.00014$ | $0.08232 \pm 0.00015$ |

Table 1: Comparison of average mean absolute error (MAE) obtained under log loss, squared loss and betting loss across varying $n/d$ and $\rho$. For each configuration, the reported value corresponds to the average MAE over repeated trials, and the value after the $\pm$ denotes the standard error.

which is an adversarial (i.e., nonstochastic) setting. In the same setting, Cesa-Bianchi et al. (2007) developed a second-order bound with the prod algorithm. Note that the notion of second-order can be defined in various ways; e.g., Hazan and Kale (2010). In $K$-armed bandits, Stoltz (2005) and Allenberg et al. (2006) have shown first-order bound. In linear bandits, obtaining a first-order regret was an open problem (Agarwal et al., 2017), which was later resolved by Allen-Zhu et al. (2018). Second-order bounds were developed by Hazan and Kale (2011) and improved by Ito et al. (2020). We refer to (Neu) for a review of the first-/second-order bounds in adversarial settings.

**Stochastic bandits with function approximation.** We now discuss first-/second-order bounds in the stochastic bandit problem with function approximation (also known as structured bandits). Hereafter, unless noted otherwise, the noise model is such that the reward (label) is bounded with a known range, which can be easily translated to $[0, 1]$-valued reward. The first-order bound was first obtained by Foster and Krishnamurthy (2021) for generic function classes. We classify second-order bounds as follows:

- **With known variance**: Based on weighted linear regression, Zhou et al. (2021); Zhou and Gu (2022); Zhao et al. (2023b) have obtained second-order bounds in linear models.

- **Unknown variances but with models of variance or distribution**: In the pure exploration setting, Weltz et al. (2023) have considered modeling the variance explicitly with a specific function class in order to obtain improved sample complexity. Wang et al. (2024a) have shown that modeling not just mean or variance but the noise distribution itself leads to a second-order bound. However, note that modeling variance or distribution has a price to pay due to the extra modeling.

- **Unknown variances**: The last set of works do not make any effort in modeling the variance or distribution, and thus there is no extra price to pay, at least in the statistical sense. For the linear model, Zhang et al. (2022) proposed a second-order regret bound, which was further improved by Kim et al. (2022). The optimal rate in this setting was first obtained by Zhao et al. (2023a), and Jun and Kim (2024) obtained the same bound but with improved numerical performance along with removal of an unnatural technical assumption on the noise. For generic function class, Jia et al. (2024) and Pacchiano (2025) both independently developed a second-order bound where the dependence of the function class appears as the eluder dimension (Russo and Van Roy, 2013).

While the work with unknown variances are the closest to our work, we emphasize that the tools developed therein do not directly imply any meaningful result for the regression setting, to our knowledge. Delineating the challenges is left as future work. That said, we believe the estimator might be useful in obtaining an improved second-order regret bound in bandits with general function classes, just in the same way that the log loss has played a role in obtaining a first-order bound (Foster and Krishnamurthy, 2021).

There is another set of work that considers sub-Gaussian noise, which is more general than the bounded reward. Kirschner and Krause (2018) consider the heteroscedastic noise in linear bandits

for the first time, to our knowledge. Their work assumes that the noise is $\sigma^2(x)$-sub-Gaussian when pulling arm $x$ and that the value of $\sigma^2(x)$ is known to the algorithm. Jun and Kim (2024) considered a further generalized setting where the noise is $\sigma_t^2$-sub-Gaussian at time step $t$, and $\sigma_t^2$ can be dependent on anything that happened up to choosing the arm $x_t$ at time $t$. Furthermore, they assume that the algorithm does not have access to $\sigma_t^2$ but rather an upper bound $\sigma_0^2$ and have shown that there exists a computationally efficient algorithm whose performance provably adapts to $\max_t \sigma_t^2$ for the leading term (though there is a lower order term with a $\sigma_0^2$ dependence).

## 6 CONCLUSION

We have introduced a new approach to regression that achieves second-order generalization guarantees by minimizing a novel *betting loss* function inspired by the betting-based confidence bounds. Our analysis establishes that minimizing this loss yields estimators whose guarantee adapts to the conditional variance of the data – without requiring any prior knowledge. Our bound is first-of-its-kind, to our knowledge.

We further demonstrate that our generalization error bounds scale favorably with the local noise level and apply broadly across both finite and infinite hypothesis classes, with a concrete instantiation for the linear function class. These results show that, under suitable conditions, variance adaptivity can be attained "for free" in the statistical sense, through a carefully chosen loss function alone and without adding extra assumptions. Despite its current computational challenges, the betting loss offers a principled framework with strong theoretical guarantees, particularly in capturing variance-adaptive behavior. Its foundational role in advancing our understanding of adaptive learning justifies further investigation, potentially inspiring new algorithmic approaches or practical surrogates.

Our results demonstrate that variance-aware learning can be achieved through the design of the loss function itself – without requiring variance estimation or modeling. This insight suggests several promising directions for extending the betting loss framework to other domains where adapting to noise is critical, such as active learning and exploration in reinforcement learning.

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

# Appendix

## Table of Contents

## A   PROOF OF THEOREM 1

**Theorem 7** (Restatement of Theorem 1). *Under log loss, we define:*

$$L_n^{\log}(f) := \sum_{(x,y) \in D_n} y \ln(\frac{1}{f_x}) + (1-y) \ln(\frac{1}{1-f_x}).$$

*Let $\hat{f} = \arg\min_{f \in \mathcal{F}} L_n^{\log}(f)$.*

*Then, with probability at least $1 - \delta$,*

$$\mathbb{E}_x |\hat{f}_x - f_x^*| \le 8\sqrt{\mathbb{E}[f_x^*(1 - f_x^*)] \frac{\ln(|\mathcal{F}|/\delta)}{n}} + 4\frac{\ln(|\mathcal{F}|/\delta)}{n} \ .$$

*Proof.* For any $f \in \mathcal{F}$, define

$$H(f) := \frac{1}{2}(L_n^{\log}(f^*) - L_n^{\log}(f)) = \sum_{(x,y) \in D_n} \frac{1}{2} y \ln(\frac{f_x}{f_x^*}) + \frac{1}{2}(1-y) \ln(\frac{1 - f_x}{1 - f_x^*}) \ .$$

Inspired by Foster and Krishnamurthy (2021), for a fixed $f \in \mathcal{F}$, consider the martingale of

$$\frac{\exp(H(f))}{\mathbb{E}[\exp(H(f))]}$$

and apply Markov's inequality to obtain that

$$1 - \delta \le \mathbb{P}\left(\frac{1}{n}H(f) \le \frac{1}{n}\ln(\mathbb{E}[\exp(H(f))]) + \frac{\ln(1/\delta)}{n}\right) \ .$$

Taking a union bound over $f \in \mathcal{F}$,

$$1 - \delta \le \mathbb{P}\left(\forall f \in \mathcal{F}, \ \frac{1}{n}H(f) \le \frac{1}{n}\ln(\mathbb{E}[\exp(H(f))]) + \frac{\ln(|\mathcal{F}|/\delta)}{n}\right) \ .$$

The rest of the proof conditions on the event that

$$\forall f \in \mathcal{F}, \ \frac{1}{n}H(f) \le \frac{1}{n}\ln(\mathbb{E}[\exp(H(f))]) + \frac{\ln(|\mathcal{F}|/\delta)}{n}.$$

By the definition of $\hat{f}$, $H(\hat{f}) \geq 0$, which implies that

$$0 \leq \frac{1}{n} \ln(\mathbb{E}[\exp(H(\hat{f}))]) + \frac{\ln(|\mathcal{F}|/\delta)}{n}. \tag{4}$$

Next, we upper bound $\frac{1}{n} \ln(\mathbb{E}[\exp(H(\hat{f}))])$.

$$\frac{1}{n} \ln(\mathbb{E}[\exp(H(\hat{f}))]) = \frac{1}{n} \ln \left( \mathbb{E}[\prod_{(x,y) \in D_n} (\frac{\hat{f}_x}{f_x^*})^{\frac{1}{2}y}(\frac{1-\hat{f}_x}{1-f_x^*})^{\frac{1}{2}(1-y)}] \right)$$

$$= \ln \left( \mathbb{E}[(\frac{\hat{f}_x}{f_x^*})^{\frac{1}{2}y}(\frac{1-\hat{f}_x}{1-f_x^*})^{\frac{1}{2}(1-y)}] \right) \qquad \text{(independence)}$$

$$\mathbb{E}[(\frac{\hat{f}_x}{f_x^*})^{\frac{1}{2}y}(\frac{1-\hat{f}_x}{1-f_x^*})^{\frac{1}{2}(1-y)}] = \mathbb{E} \exp \left( \frac{1}{2}y \ln(\frac{\hat{f}_x}{f_x^*}) + \frac{1}{2}(1-y) \ln(\frac{1-\hat{f}_x}{1-f_x^*}) \right)$$

$$= \mathbb{E} \exp \left( \mathbb{E}' \frac{1}{2}y' \ln(\frac{\hat{f}_x}{f_x^*}) + \frac{1}{2}(1-y') \ln(\frac{1-\hat{f}_x}{1-f_x^*}) \right)$$

$$\qquad (y' \sim \text{Bernoulli}(y))$$

$$\leq \mathbb{E}\,\mathbb{E}' \exp \left( \frac{1}{2}y' \ln(\frac{\hat{f}_x}{f_x^*}) + \frac{1}{2}(1-y') \ln(\frac{1-\hat{f}_x}{1-f_x^*}) \right)$$

$$\qquad \text{(Jensen's inequality)}$$

$$= \mathbb{E}_x\, f_x^* \cdot \sqrt{\frac{\hat{f}_x}{f_x^*}} + (1-f_x^*) \cdot \sqrt{\frac{1-\hat{f}_x}{1-f_x^*}}$$

$$= \mathbb{E}_x[f_x^* \hat{f}_x + (1-f_x^*)(1-\hat{f}_x)]$$

Combining with Eqn. (4),

$$\frac{\ln(|\mathcal{F}|/\delta)}{n} \geq -\ln(\mathbb{E}_x[f_x^* \hat{f}_x + (1-f_x^*)(1-\hat{f}_x)])$$

$$= -\ln(1 - \mathbb{E}[1 - f_x^* \hat{f}_x - (1-f_x^*)(1-\hat{f}_x)])$$

$$\geq \mathbb{E}[1 - f_x^* \hat{f}_x - (1-f_x^*)(1-\hat{f}_x)] \qquad (\ln(1+x) \leq x)$$

$$= \mathbb{E}[\frac{1}{2}(\sqrt{f_x^*} - \sqrt{\hat{f}_x})^2 + \frac{1}{2}(\sqrt{1-f_x^*} - \sqrt{1-\hat{f}_x})^2]$$

$$= \mathbb{E}[D^2(f_x^*, \hat{f}_x)] \tag{5}$$

where $D^2(p,q)$ for scalers $p, q \in [0,1]$ denotes the Hellinger distance between two Bernoulli distributions with parameters $p$ and $q$: i.e., $D^2(p,q) = \frac{1}{2}(\sqrt{p} - \sqrt{q})^2 + \frac{1}{2}(\sqrt{1-p} - \sqrt{1-q})^2$.

From the proof of Proposition 3 of Foster and Krishnamurthy (2021), we know that

$$D^2(p,q) = \frac{1}{2}(\sqrt{p} - \sqrt{q})^2 + \frac{1}{2}(\sqrt{1-p} - \sqrt{1-q})^2$$

$$= \frac{(p-q)^2}{2} \cdot \left( \frac{1}{(\sqrt{p} + \sqrt{q})^2} + \frac{1}{(\sqrt{1-p} + \sqrt{1-q})^2} \right)$$

$$\geq \frac{(p-q)^2}{2} \cdot \left( \frac{1}{(\sqrt{p} + \sqrt{q})^2 \wedge (\sqrt{1-p} + \sqrt{1-q})^2} \right)$$

$$\geq \frac{(p-q)^2}{4} \cdot \left( \frac{1}{(p+q) \wedge (1-p+1-q)} \right) \qquad ((a+b)^2 \leq 2a^2 + 2b^2)$$

Let $g(p,q) = (p+q) \wedge (1-p+1-q)$. Then, by Eqn. (5),

$$\mathbb{E} \left[ (f_x^* - \hat{f}_x)^2 \cdot \frac{1}{2g(f_x^*, \hat{f}_x)} \right] \leq 2\frac{\ln(|\mathcal{F}|/\delta)}{n}$$

Using $\frac{A^2}{2B} = \max_{\eta>0} \eta A - \frac{\eta^2}{2}B$ for $A, B > 0$, we have, for any $\eta > 0$,

$$2\frac{\ln(|\mathcal{F}|/\delta)}{n} \geq \mathbb{E}[\max_\eta \eta|f_x^* - \hat{f}_x| - \frac{\eta^2}{2}g(f_x^*, \hat{f}_x)]$$

$$\geq \max_\eta \eta\,\mathbb{E}[|f_x^* - \hat{f}_x|] - \frac{\eta^2}{2}\,\mathbb{E}[g(f_x^*, \hat{f}_x)] \qquad \text{(Jensen)}$$

$$\implies \mathbb{E}[|f_x^* - \hat{f}_x|] \leq \min_\eta \frac{\eta}{2}\,\mathbb{E}[g(f_x^*, \hat{f}_x)] + \frac{1}{\eta}\frac{2\ln(|\mathcal{F}|/\delta)}{n}\,.$$

Note that

$$\mathbb{E}\,g(f_x^*, \hat{f}_x) = \mathbb{E}[(f_x^* + \hat{f}_x) \wedge (1 - f_x^* + 1 - \hat{f}_x)]$$

$$\leq \mathbb{E}[(|f_x^* - \hat{f}_x| + 2f_x^*) \wedge (|f_x^* - \hat{f}_x| + 2(1 - f_x^*))]$$

$$= \mathbb{E}[|f_x^* - \hat{f}_x| + (2f_x^* \wedge 2(1 - f_x^*))]$$

$$= \mathbb{E}[|f_x^* - \hat{f}_x|] + 2\,\mathbb{E}[f_x^* \wedge (1 - f_x^*)]$$

$$\leq \mathbb{E}[|f_x^* - \hat{f}_x|] + 4\,\mathbb{E}[f_x^*(1 - f_x^*)].$$

Then,

$$\mathbb{E}[|f_x^* - \hat{f}_x|] \leq \frac{\eta}{2}\,\mathbb{E}[|f_x^* - \hat{f}_x|] + 2\eta\,\mathbb{E}[f_x^*(1 - f_x^*)] + \frac{1}{\eta}\frac{2\ln(|\mathcal{F}|/\delta)}{n}$$

$$\leq \frac{1}{2}\,\mathbb{E}[|f_x^* - \hat{f}_x|] + 2\eta\,\mathbb{E}[f_x^*(1 - f_x^*)] + \frac{1}{\eta}\frac{2\ln(|\mathcal{F}|/\delta)}{n} \qquad \text{(assume } \eta \leq 1\text{)}$$

$$\implies \mathbb{E}[|f_x^* - \hat{f}_x|] \leq 4\eta\,\mathbb{E}[f_x^*(1 - f_x^*)] + \frac{4}{\eta}\frac{\ln(|\mathcal{F}|/\delta)}{n}$$

We can choose $\eta = 1 \wedge \sqrt{\frac{\ln(|\mathcal{F}|/\delta)/n}{\mathbb{E}[f_x^*(1-f_x^*)]}}$, which satisfies the assumption above, to arrive at

$$\mathbb{E}[|f_x^* - \hat{f}_x|] \leq 8\sqrt{\mathbb{E}[f_x^*(1 - f_x^*)]\frac{\ln(|\mathcal{F}|/\delta)}{n}} + 4\frac{\ln(|\mathcal{F}|/\delta)}{n}\,.$$

$\square$

## B  PROOF OF THEOREM 3

**Definition 8.** *We first provide definitions for new quantities that are used throughout the proof of Theorem 3.*

$$\Delta_x := f_x^* - f_x$$

$$\overline{\Delta}_{h,x,\phi,c} := \overline{(\phi(h_x - f_x))}_{[-c,c]}$$

$$U_x := \max\{(-f_x^*)\frac{-\overline{\Delta}_{h,x,\phi,c}}{1 + \Delta_x\overline{\Delta}_{h,x,\phi,c}}, \; (1 - f_x^*)\frac{-\overline{\Delta}_{h,x,\phi,c}}{1 + \Delta_x\overline{\Delta}_{h,x,\phi,c}}\}$$

**Lemma 9.** *For any $x \in \mathcal{X}$, we have:*

1. $\overline{\Delta}_{f^*,x,\phi,c} = \mathrm{sign}(f_x^* - f_x)\,(\phi|f_x^* - f_x| \wedge c)$, *and* $\Delta_x\overline{\Delta}_{f^*,x,\phi,c} \geq 0$.

2. $U_x \leq \frac{1}{4}$.

*Proof.* 1. By the definition of $\overline{\Delta}_{h,x,\phi,c}$, one can see $\overline{\Delta}_{f^*,x,\phi,c} = \overline{(\phi(f_x^* - f_x))}_{[-c,c]} = \mathrm{sign}(f_x^* - f_x)\,(\phi|f_x^* - f_x| \wedge c)$, and $\Delta_x\overline{\Delta}_{f^*,x,\phi,c} \geq 0$ for all $x$.

2. Note that

$$|\overline{\Delta}_{f^*,x,\phi,c}| = \phi|f_x^* - f_x| \wedge c$$

$$\leq c$$
$$\leq \frac{1}{4} \qquad\qquad (c \leq \tfrac{1}{4})$$

If $\overline{\Delta}_{f^*,x,\phi,c} \geq 0$,

$$U_x = \max\{(-f_x^*)\frac{-\overline{\Delta}_{f^*,x,\phi,c}}{1+\Delta_x\overline{\Delta}_{f^*,x,\phi,c}},\ (1-f_x^*)\frac{-\overline{\Delta}_{f^*,x,\phi,c}}{1+\Delta_x\overline{\Delta}_{f^*,x,\phi,c}}\}$$

$$= f_x^*\frac{\overline{\Delta}_{f^*,x,\phi,c}}{1+\Delta_x\overline{\Delta}_{f^*,x,\phi,c}}$$

$$\leq \overline{\Delta}_{f^*,x,\phi,c} \qquad\qquad (\forall x,\ \Delta_x\overline{\Delta}_{f^*,x,\phi,c} \geq 0,\ 0 \leq f_x^* \leq 1)$$

$$\leq \frac{1}{4} \qquad\qquad (|\overline{\Delta}_{f^*,x,\phi,c}| \leq \tfrac{1}{4})$$

Similarly, we can show that if $\overline{\Delta}_{f^*,x,\phi,c} \leq 0$, then $U_x = (1-f_x^*)\frac{-\overline{\Delta}_{f^*,x,\phi,c}}{1+\Delta_x\overline{\Delta}_{f^*,x,\phi,c}} \leq \frac{1}{4}$.

$\square$

**Lemma 10.** *Let $a \in (0,1)$. Then, $\forall x \in [0,a]$, $\ln(1-x) \geq \frac{-\ln(1-a)}{a} \cdot (-x)$.*

*Proof.* Given the concavity of $\ln(1-x)$, for any $x \in [0,a]$, the function lies above the secant line connecting $(0, \ln(1-0))$ and $(a, \ln(1-a))$.

The equation of the secant line is:
$$y = \frac{\ln(1-a)}{a}x.$$

By concavity:
$$\ln(1-x) \geq \frac{\ln(1-a)}{a}x.$$

$\square$

**Lemma 11.** *Let $\delta \in (0, \frac{1}{|\mathcal{F}|})$. We have,*

$$1 - |\mathcal{F}|\delta \leq \mathbb{P}\left(\forall h \in \mathcal{F},\ \phi \in [0,\overline{\phi}],\ c \in [0,\tfrac{1}{4}],\ \frac{1}{n}H_{\phi,c}(h,f^*) \leq \frac{1}{n}\ln(8\overline{\phi}n^2/\delta)\right)$$

*Proof.* The plan is to fix $h$ and show

$$1 - \delta \leq \mathbb{P}\left(\forall \phi^* \in [0,\overline{\phi}], c^* \in [0,\tfrac{1}{4}],\ \frac{1}{n}H_{\phi^*,c^*}(h,f^*) \leq \frac{1}{n}\ln(8\overline{\phi}n^2/\delta)\right)$$

and then take the union bound over $h \in \mathcal{F}$.

Let $\varepsilon > 0$ be a small number to be chosen later. Discretize $[0,\overline{\phi}] \times [0,\tfrac{1}{4}]$ as blocks of length $\varepsilon$ by $\varepsilon$. The number of such blocks is $\frac{\overline{\phi}}{4\varepsilon^2}$. For any $(\phi^*, c^*)$, there is block, such that $(\phi^*, c^*)$ belongs to this block. Let $U'$ be the uniform distribution supported on this block.

We start from the martingale

$$\frac{\mathbb{E}_{(\phi,c)\sim U'}[\exp(H_{\phi,c}(h,f^*))]}{\mathbb{E}_{(\phi,c)\sim U',\{x,y\}\sim D^n}[\exp(H_{\phi,c}(h,f^*))]}$$

Using Markov's inequality, we have, w.p. at least $1 - \delta/(\frac{\overline{\phi}}{4\varepsilon^2})$,

$$\ln(\mathbb{E}_{(\phi,c)\sim U'}[\exp(H_{\phi,c}(h,f^*))]) \leq \ln(\mathbb{E}_{(\phi,c)\sim U',\{(x,y)\}\sim D^n}[\exp(H_{\phi,c}(h,f^*))]) + \ln(\frac{\overline{\phi}}{4\varepsilon^2\delta})$$

$$= \ln(\mathbb{E}_{(\phi,c)\sim U'}(\mathbb{E}_{\{(x,y)\}\sim D}[1 + (y-f^*)\overline{(\phi(h_x - f_x))}_{[-c,c]}])^n) + \ln(\frac{\overline{\phi}}{4\varepsilon^2\delta})$$

(independence)

$$= \ln(\frac{\overline{\phi}}{4\varepsilon^2\delta}) \tag{6}$$

Taking a union bound over all $\frac{\overline{\phi}}{4\varepsilon^2}$ blocks, we have with probability at least $1 - \delta$, for any $U$ that is a uniform distribution on any block,

$$\ln(\mathbb{E}_{(\phi,c)\sim U}[\exp(H_{\phi,c}(h, f^*))]) \leq \ln(\frac{\overline{\phi}}{4\varepsilon^2\delta})$$

We desire to lower bound the LHS of Equation (6) above as $H_{\phi^*,c^*}(h, f^*)$ plus some extra terms for any $(\phi^*, c^*)$ that belongs to the support of $U'$.

Note that

$$\mathbb{E}_{(\phi,c)\sim U'}[\exp(H_{\phi,c}(h, f^*))] = \mathbb{E}_{(\phi,c)\sim U'}\left(\prod_{(x,y)}\left(1 + (y - f_x^*)\overline{(\phi(h_x - f_x^*))}_{[-c,c]}\right)\right).$$

Note that if $|\phi^* - \phi| \leq \varepsilon$ and $|c^* - c| \leq \varepsilon$, then using 1-Lipschitzness of $F_1(\phi) = 1 + (y - f_x^*)\overline{(\phi(h_x - f_x^*))}_{[-c,c]}$, and 1-Lipschitzness of $F_2(c) = 1 + (y - f_x^*)\overline{(\phi(h_x - f_x^*))}_{[-c,c]}$,

$$1 + (y - f_x^*)\overline{(\phi(h_x - f_x^*))}_{[-c,c]}$$
$$\geq 1 + (y - f_x^*)\overline{(\phi^*(h_x - f_x^*))}_{[-c,c]} - \varepsilon$$
$$\geq 1 + (y - f_x^*)\overline{(\phi^*(h_x - f_x^*))}_{[-c^*,c^*]} - 2\varepsilon$$
$$= (1 + (y - f_x^*)\overline{(\phi^*(h_x - f_x^*))}_{[-c^*,c^*]}) \cdot (1 - \frac{2\varepsilon}{1 + (y - f_x^*)\overline{(\phi^*(h_x - f_x^*))}_{[-c^*,c^*]}})$$
$$\geq (1 + (y - f_x^*)\overline{(\phi^*(h_x - f_x^*))}_{[-c^*,c^*]}) \cdot (1 - \frac{8}{3}\varepsilon) \tag{$c^* \leq \frac{1}{4}$}$$

Thus,

$$\ln(\mathbb{E}_{(\phi,c)\sim U}[\exp(H_{\phi,c}(h, f^*))]) \geq \sum_{(x,y)}\ln\left(1 + (y - f_x^*)\overline{(\phi^*(h_x - f_x^*))}_{[-c^*,c^*]}\right) + n\ln(1 - \frac{8}{3}\varepsilon)$$
$$\geq \sum_{(x,y)}\ln\left(1 + (y - f_x^*)\overline{(\phi^*(h_x - f_x^*))}_{[-c^*,c^*]}\right) - n\varepsilon$$
$$\text{(Lemma 10; } \varepsilon \leq \frac{1}{8})$$

This implies that

$$\frac{1}{n}H_{\phi^*,c^*}(h, f^*) \leq \varepsilon + \frac{1}{n}\ln(\frac{\overline{\phi}}{4\varepsilon^2\delta})$$

Choosing $\varepsilon = \frac{1}{4n}$, the RHS of above inequality can be upper bounded as:

$$\varepsilon + \frac{1}{n}\ln(\frac{\overline{\phi}}{4\varepsilon^2\delta}) \leq \frac{1}{n}\ln(8\overline{\phi}n^2/\delta)$$

concluding the proof. $\qquad\square$

**Lemma 12.** *Let $\delta \in (0, \frac{1}{|\mathcal{F}|})$. Then,*

$$1 - |\mathcal{F}|\delta \leq \mathbb{P}\left(\forall f \in \mathcal{F},\ \phi \in [0, \overline{\phi}],\ c \in [0, \frac{1}{4}],\right.$$

$$\left. -\frac{1}{n}H_{\phi,c}(f^*, f) \leq \mathbb{E}_x\left[-\frac{\Delta_x\overline{\Delta}_{f^*,x,\phi,c}}{1 + \Delta_x\overline{\Delta}_{f^*,x,\phi,c}} + \frac{4}{3}\cdot\sigma_x^2\overline{\Delta}_{f^*,x,\phi,c}^2\right] + \frac{1}{n}\ln(24\overline{\phi}n^2/\delta)\right)$$

*Proof.* The plan is to fix $f$ and show

$$1 - \delta \leq \mathbb{P}\left(\forall \phi^* \in [0, \overline{\phi}], c^* \in [0, \frac{1}{4}],\ -\frac{1}{n}H_{\phi^*,c^*}(f^*, f)\right.$$

$$\leq \mathbb{E}_x \left[ -\frac{\Delta_x \overline{\Delta}_{f^*,x,\phi^*,c^*}}{1 + \Delta_x \overline{\Delta}_{f^*,x,\phi^*,c^*}} + \frac{4}{3} \cdot \sigma_x^2 \overline{\Delta}^2_{f^*,x,\phi^*,c^*} \right] + \frac{1}{n} \ln(24\overline{\phi}n^2/\delta) \Bigg)$$

and then take the union bound over $f \in \mathcal{F}$.

Let $\varepsilon > 0$ be a small number to be chosen later. Discretize $[0, \overline{\phi}] \times [0, \frac{1}{4}]$ as blocks of length $\varepsilon$ by $\varepsilon$. The number of such blocks is $\frac{\overline{\phi}}{4\varepsilon^2}$. For any $(\phi^*, c^*)$, there is block, such that $(\phi^*, c^*)$ belongs to this block. Let $U'$ be the uniform distribution supported on this block.

We start from the martingale

$$\frac{\mathbb{E}_{(\phi,c)\sim U'}[\exp(-H_{\phi,c}(f^*, f))]}{\mathbb{E}_{(\phi,c)\sim U', \{x,y\}\sim D}[\exp(-H_{\phi,c}(f^*, f))]}$$

Using Markov's inequality, we have, w.p. at least $1 - \delta/(\frac{\overline{\phi}}{4\varepsilon^2})$,

$$\ln(\mathbb{E}_{(\phi,c)\sim U'}[\exp(-H_{\phi,c}(f^*, f))])$$

$$\leq \ln(\mathbb{E}_{(\phi,c)\sim U', \{(x,y)\}\sim D^n}[\exp(-H_{\phi,c}(f^*, f))]) + \ln(\frac{\overline{\phi}}{4\varepsilon^2}/\delta)$$

$$\leq n \, \mathbb{E}_{(\phi,c)\sim U'} \, \mathbb{E}_x \left[ -\frac{\Delta_x \overline{\Delta}_{f^*,x,\phi,c}}{1 + \Delta_x \overline{\Delta}_{f^*,x,\phi,c}} + \frac{1}{(1 + \Delta_x \overline{\Delta}_{f^*,x,\phi,c})^3(1 - U_x)} \cdot \sigma_x^2 \overline{\Delta}^2_{f^*,x,\phi,c} \right] + \ln(\frac{\overline{\phi}}{4\varepsilon^2}/\delta)$$

$$\tag{7}$$

where the last inequality is by Lemma 13.

Taking a union bound over all $\frac{\overline{\phi}}{4\varepsilon^2}$ blocks, we have with probability at least $1 - \delta$, for any $U$ that is a uniform distribution on any block,

$$\ln(\mathbb{E}_{(\phi,c)\sim U}[\exp(-H_{\phi,c}(f^*, f))])$$

$$\leq n \, \mathbb{E}_{(\phi,c)\sim U'} \, \mathbb{E}_x \left[ -\frac{\Delta_x \overline{\Delta}_{f^*,x,\phi,c}}{1 + \Delta_x \overline{\Delta}_{f^*,x,\phi,c}} + \frac{1}{(1 + \Delta_x \overline{\Delta}_{f^*,x,\phi,c})^3(1 - U_x)} \cdot \sigma_x^2 \overline{\Delta}^2_{f^*,x,\phi,c} \right] + \ln(\frac{\overline{\phi}}{4\varepsilon^2}/\delta)$$

We upper bound the RHS of (7) as follows:

$$\mathbb{E}_{(\phi,c)\sim U'} \, \mathbb{E}_x \left[ -\frac{\Delta_x \overline{\Delta}_{f^*,x,\phi,c}}{1 + \Delta_x \overline{\Delta}_{f^*,x,\phi,c}} + \frac{1}{(1 + \Delta_x \overline{\Delta}_{f^*,x,\phi,c})^3(1 - U_x)} \cdot \sigma_x^2 \overline{\Delta}^2_{f^*,x,\phi,c} \right]$$

$$\leq \mathbb{E}_{(\phi,c)\sim U'} \, \mathbb{E}_x \left[ -\frac{\Delta_x \overline{\Delta}_{f^*,x,\phi,c}}{1 + \Delta_x \overline{\Delta}_{f^*,x,\phi,c}} + \frac{4}{3} \cdot \sigma_x^2 \overline{\Delta}^2_{f^*,x,\phi,c} \right] \text{ (By Lemma 9: } \Delta_x \overline{\Delta}_{g,x,c} \geq 0, \ U_x \leq \frac{1}{4})$$

One can see that $\overline{\Delta}_{f^*,x,\phi,c} = \overline{(\phi(f_x^* - f_x))}_{[-c,c]}$ is 1-Lipschitz in $\phi$ and 1-Lipschitz in $c$, i.e., $F_1(\phi) = \overline{\Delta}_{f^*,x,\phi,c}$ is 1-Lipschitz, and $F_2(c) = \overline{\Delta}_{f^*,x,\phi,c}$ is 1-Lipschitz. Further, $F_1^2(\phi)$ and $F_2^2(c)$ are 1-Lipschitz since $\overline{\Delta}_{f^*,x,\phi,c} \leq c \leq \frac{1}{4}$. In addition, since for $x \in [0, \frac{1}{4}], |\frac{\mathrm{d}}{\mathrm{d}x} \frac{x}{1+x}| = \frac{1}{(1+x)^2} \leq 1$, $\frac{\Delta_x \overline{\Delta}_{f^*,x,\phi,c}}{1 + \Delta_x \overline{\Delta}_{f^*,x,\phi,c}}$ is 1-Lipschitz in $\phi$ and 1-Lipschitz in $c$.

Note that if $|\phi^* - \phi| \leq \varepsilon$ and $|c^* - c| \leq \varepsilon$, then using Lipchitzness arguments above, as well as $\sigma_x^2 \leq \frac{1}{4}, \forall x$, we have

$$-\frac{\Delta_x \overline{\Delta}_{f^*,x,\phi,c}}{1 + \Delta_x \overline{\Delta}_{f^*,x,\phi,c}} + \frac{4}{3} \cdot \sigma_x^2 \overline{\Delta}^2_{f^*,x,\phi,c} \leq -\frac{\Delta_x \overline{\Delta}_{f^*,x,\phi^*,c}}{1 + \Delta_x \overline{\Delta}_{f^*,x,\phi^*,c}} + \frac{4}{3} \cdot \sigma_x^2 \overline{\Delta}^2_{f^*,x,\phi^*,c} + 2\varepsilon$$

$$\leq -\frac{\Delta_x \overline{\Delta}_{f^*,x,\phi^*,c^*}}{1 + \Delta_x \overline{\Delta}_{f^*,x,\phi^*,c^*}} + \frac{4}{3} \cdot \sigma_x^2 \overline{\Delta}^2_{f^*,x,\phi^*,c^*} + 4\varepsilon$$

This implies that,

$$\mathbb{E}_{(\phi,c)\sim U'} \, \mathbb{E}_x \left[ -\frac{\Delta_x \overline{\Delta}_{f^*,x,\phi,c}}{1 + \Delta_x \overline{\Delta}_{f^*,x,\phi,c}} + \frac{4}{3} \cdot \sigma_x^2 \overline{\Delta}^2_{f^*,x,\phi,c} \right]$$

$$\leq \mathbb{E}_x\left[-\frac{\Delta_x\overline{\Delta}_{f^*,x,\phi^*,c^*}}{1+\Delta_x\overline{\Delta}_{f^*,x,\phi^*,c^*}}+\frac{4}{3}\cdot\sigma_x^2\overline{\Delta}_{f^*,x,\phi^*,c^*}^2\right]+4\varepsilon$$

For the LHS of (7),

$$\mathbb{E}_{(\phi,c)\sim U'}[\exp(-H_{\phi,c}(f^*,f))]=\mathbb{E}_{(\phi,c)\sim U'}\left(\prod_{(x,y)}\frac{1}{\left(1+(y-f_x)\overline{(\phi(f_x^*-f_x))}_{[-c,c]}\right)}\right).$$

Note that if $|\phi^*-\phi|\leq\varepsilon$ and $|c^*-c|\leq\varepsilon$, then using 1-Lipschitzness of $F_3(\phi)=1+(y-f_x)\overline{(\phi(f_x^*-f_x))}_{[-c,c]}$, and 1-Lipschitzness of $F_4(c)=1+(y-f_x)\overline{(\phi(f_x^*-f_x))}_{[-c,c]}$,

$$1+(y-f_x)\overline{(\phi(f_x^*-f_x))}_{[-c,c]}$$
$$\leq 1+(y-f_x)\overline{(\phi^*(f_x^*-f_x))}_{[-c,c]}+\varepsilon$$
$$\leq 1+(y-f_x)\overline{(\phi^*(f_x^*-f_x))}_{[-c^*,c^*]}+2\varepsilon$$
$$=(1+(y-f_x)\overline{(\phi^*(f_x^*-f_x))}_{[-c^*,c^*]})\cdot(1+\frac{2\varepsilon}{1+(y-f_x)\overline{(\phi^*(f_x^*-f_x))}_{[-c^*,c^*]}})$$
$$\leq (1+(y-f_x)\overline{(\phi^*(f_x^*-f_x))}_{[-c^*,c^*]})\cdot(1+\frac{8}{3}\varepsilon) \qquad (c^*\leq\frac{1}{4})$$

Thus,

$$\ln\left(\mathbb{E}_{(\phi,c)\sim U'}[\exp(-H_{\phi,c}(f^*,f))]\right)=\mathbb{E}_{(\phi,c)\sim U'}\left(\prod_{(x,y)}\frac{1}{\left(1+(y-f_x)\overline{(\phi(f_x^*-f_x))}_{[-c,c]}\right)}\right)$$
$$\geq \sum_{(x,y)}-\ln\left(1+(y-f_x)\overline{(\phi^*(f_x^*-f_x))}_{[-c^*,c^*]}\right)-n\ln(1+\frac{8}{3}\varepsilon)$$
$$\geq \sum_{(x,y)}-\ln\left(1+(y-f_x)\overline{(\phi^*(f_x^*-f_x))}_{[-c^*,c^*]}\right)-n\frac{8}{3}\varepsilon$$
$$(\ln(1+x)\leq x)$$

Combining the bounds for the LHS and RHS of (7),

$$\sum_{(x,y)}-\ln\left(1+(y-f_x)\overline{(\phi^*(f_x^*-f_x))}_{[-c^*,c^*]}\right)-n\frac{8}{3}\varepsilon$$
$$\leq n\,\mathbb{E}_x\left[-\frac{\Delta_x\overline{\Delta}_{f^*,x,\phi^*,c^*}}{1+\Delta_x\overline{\Delta}_{f^*,x,\phi^*,c^*}}+\frac{4}{3}\sigma_x^2\overline{\Delta}_{f^*,x,\phi^*,c^*}^2\right]+4n\varepsilon+\ln(\frac{\overline{\phi}}{4\varepsilon^2}/\delta)$$

This implies that

$$-\frac{1}{n}H_{\phi^*,c^*}(f^*,f)\leq\frac{20}{3}\varepsilon+\mathbb{E}_x\left[-\frac{\Delta_x\overline{\Delta}_{f^*,x,\phi^*,c^*}}{1+\Delta_x\overline{\Delta}_{f^*,x,\phi^*,c^*}}+\frac{4}{3}\cdot\sigma_x^2\overline{\Delta}_{f^*,x,\phi^*,c^*}^2\right]+\frac{1}{n}\ln(\frac{\overline{\phi}}{4\varepsilon^2}/\delta)$$

Choosing $\varepsilon=\frac{1}{4n}$,

$$-\frac{1}{n}H_{\phi^*,c^*}(f^*,f)\leq\frac{20}{3}\varepsilon+\mathbb{E}_x\left[-\frac{\Delta_x\overline{\Delta}_{f^*,x,\phi^*,c^*}}{1+\Delta_x\overline{\Delta}_{f^*,x,\phi^*,c^*}}+\frac{4}{3}\cdot\sigma_x^2\overline{\Delta}_{f^*,x,\phi^*,c^*}^2\right]+\frac{1}{n}\ln(\frac{\overline{\phi}}{4\varepsilon^2}/\delta)$$
$$\leq \mathbb{E}_x\left[-\frac{\Delta_x\overline{\Delta}_{f^*,x,\phi^*,c^*}}{1+\Delta_x\overline{\Delta}_{f^*,x,\phi^*,c^*}}+\frac{4}{3}\cdot\sigma_x^2\overline{\Delta}_{f^*,x,\phi^*,c^*}^2\right]+\frac{1}{n}\ln(24\overline{\phi}n^2/\delta)$$

$\square$

**Lemma 13.** *Recall the definition of the loss function $H_{\phi,c}$ and $U_x = \max\{(-f_x^*)\frac{-\overline{\Delta}_{h,x,\phi,c}}{1+\Delta_x\overline{\Delta}_{h,x,\phi,c}},\ (1-$*
*$f_x^*)\frac{-\overline{\Delta}_{h,x,\phi,c}}{1+\Delta_x\overline{\Delta}_{h,x,\phi,c}}\}$. Let $V$ be a distribution of $(\phi,c)$ supported on a subset of $[0,\overline{\phi}]\times[0,\frac{1}{4}]$. Then for any $h,f \in \mathcal{F}$, we have*

$$\ln(\mathbb{E}_{(\phi,c)\sim V,\{(x,y)\}\sim D^n}[\exp(-H_{\phi,c}(h,f))])$$

$$\leq n\,\mathbb{E}_{(\phi,c)\sim V}\,\mathbb{E}_x\left[-\frac{\Delta_x\overline{\Delta}_{h,x,\phi,c}}{1+\Delta_x\overline{\Delta}_{h,x,\phi,c}} + \frac{1}{(1+\Delta_x\overline{\Delta}_{h,x,\phi,c})^3(1-U_x)}\cdot\sigma_x^2\overline{\Delta}_{h,x,\phi,c}^2\right]$$

*Proof.* Let $\eta := y - f^*$, then $\forall x \in \mathcal{X}$, $\mathbb{E}[\eta \mid x] = 0$ and $\mathbb{E}[\eta^2 \mid x] = \sigma_x^2$. We have

$$\left(\mathbb{E}_{(\phi,c)\sim V,\{(x,y)\}\sim D^n}[\exp(-H_{\phi,c}(h,f))]\right)^{\frac{1}{n}}$$

$$= \mathbb{E}_{(\phi,c)\sim V,\{(x,y)\}\sim D}\left[\left(\frac{1}{1+(y-f_x)\overline{(\phi(h_x-f_x))}_{[-c,c]}}\right)\right]$$

$$= \mathbb{E}_{(\phi,c)\sim V,\{(x,y)\}\sim D}\left[\left(\frac{1}{1+(f_x^*+\eta-f_x)\overline{(\phi(h_x-f_x))}_{[-c,c]}}\right)\right]$$

$$= \mathbb{E}_{(\phi,c)\sim V,\{(x,y)\}\sim D}\left[\left(\frac{1}{1+\Delta_x\overline{\Delta}_{h,x,\phi,c}+\eta\overline{\Delta}_{h,x,\phi,c}}\right)\right]$$

$$= \mathbb{E}_{(\phi,c)\sim V}\,\mathbb{E}_x\left[\frac{1}{1+\Delta_x\overline{\Delta}_{h,x,\phi,c}}\,\mathbb{E}_\eta\left[\frac{1}{1+\eta\overline{\Delta}_{h,x,\phi,c}\cdot(1+\Delta_x\overline{\Delta}_{h,x,\phi,c})^{-1}}\right]\right]$$

Using the fact that $\frac{1}{1+x} = 1 - x + \frac{x^2}{1+x}$ with $x = \eta\overline{\Delta}_{h,x,\phi,c}\cdot(1+\Delta_x\overline{\Delta}_{h,x,\phi,c})^{-1}$, we have

$$\mathbb{E}_\eta\left[\frac{1}{1+\eta\overline{\Delta}_{h,x,\phi,c}\cdot(1+\Delta_x\overline{\Delta}_{h,x,\phi,c})^{-1}}\right] = 1 + \mathbb{E}_\eta\left[\frac{\eta^2\overline{\Delta}_{h,x,\phi,c}^2}{(1+\Delta_x\overline{\Delta}_{h,x,\phi,c})^2}\cdot\frac{1}{1+\eta\overline{\Delta}_{h,x,\phi,c}\cdot(1+\Delta_x\overline{\Delta}_{h,x,\phi,c})^{-1}}\right]$$

If $\overline{\Delta}_{h,x,\phi,c} \geq 0$, then the RHS $\leq 1 + \frac{\sigma_x^2\overline{\Delta}_{h,x,\phi,c}^2}{(1+\Delta_x\overline{\Delta}_{h,x,\phi,c})^2}\cdot\frac{1}{1+(-f^*)\overline{\Delta}_{h,x,\phi,c}\cdot(1+\Delta_x\overline{\Delta}_{h,x,\phi,c})^{-1}}$. Else if $\overline{\Delta}_{h,x,\phi,c} < 0$, then the RHS $\leq 1 + \frac{\sigma_x^2\overline{\Delta}_{h,x,\phi,c}^2}{(1+\Delta_x\overline{\Delta}_{h,x,\phi,c})^2}\cdot\frac{1}{1+(1-f^*)\overline{\Delta}_{h,x,\phi,c}\cdot(1+\Delta_x\overline{\Delta}_{h,x,\phi,c})^{-1}}$.

Thus, with $U_x = \max\{(-f_x^*)\frac{-\overline{\Delta}_{h,x,\phi,c}}{1+\Delta_x\overline{\Delta}_{h,x,\phi,c}},\ (1-f_x^*)\frac{-\overline{\Delta}_{h,x,\phi,c}}{1+\Delta_x\overline{\Delta}_{h,x,\phi,c}}\}$,

$$\frac{1}{n}\ln(\mathbb{E}_{(\phi,c)\sim V,\{(x,y)\}\sim D^n}[\exp(-H_{\phi,c}(h,f))])$$

$$\leq \ln\mathbb{E}_{(\phi,c)\sim V}\,\mathbb{E}_x\left[\frac{1}{1+\Delta_x\overline{\Delta}_{h,x,\phi,c}}\left(1+\sigma_x^2\overline{\Delta}_{h,x,\phi,c}^2\cdot\frac{1}{(1+\Delta_x\overline{\Delta}_{h,x,\phi,c})^2(1-U_x)}\right)\right]$$

$$\leq \mathbb{E}_{(\phi,c)\sim V}\,\mathbb{E}_x\left[\frac{1}{1+\Delta_x\overline{\Delta}_{h,x,\phi,c}}\left(1+\sigma_x^2\overline{\Delta}_{h,x,\phi,c}^2\cdot\frac{1}{(1+\Delta_x\overline{\Delta}_{h,x,\phi,c})^2(1-U_x)}\right)-1\right]$$

$$(\ln x \leq x - 1)$$

$$= \mathbb{E}_{(\phi,c)\sim V}\,\mathbb{E}_x\left[\frac{1}{1+\Delta_x\overline{\Delta}_{h,x,\phi,c}}\left(\sigma_x^2\overline{\Delta}_{h,x,\phi,c}^2\cdot\frac{1}{(1+\Delta_x\overline{\Delta}_{h,x,\phi,c})^2(1-U_x)}-\Delta_x\overline{\Delta}_{h,x,\phi,c}\right)\right]$$

completing the proof. $\square$

**Theorem 14** (Restatement of Theorem 3). *Recall that*

$$L_n(f) := \max_{h\in\mathcal{F}}\max_{\phi\in[0,\overline{\phi}]}\max_{c\in[0,\frac{1}{4}]}\frac{1}{n}\sum_{(x,y)\in D_n}\ln\left(1+(y-f_x)\overline{(\phi(h_x-f_x))}_{[-c,c]}\right)$$

*With probability at least $1 - \delta$, $\forall f \in \mathcal{F}$,*

$$\mathbb{E}_x |f_x - f_x^*|$$

$$\leq \sqrt{\frac{25}{12} \mathbb{E}\, \sigma_x^2 \cdot \left( \frac{2}{n} \ln\left( \frac{48|\mathcal{F}|\overline{\phi}n^2}{\delta} \right) + (L_n(f) - L_n(f^*)) \right)} + \frac{6}{n} \ln\left( \frac{48|\mathcal{F}|\overline{\phi}n^2}{\delta} \right) + \frac{5}{2}(L_n(f) - L_n(f^*))$$

*Proof.* Define the events

$$A_1 := \left\{ \forall h \in \mathcal{F},\ \phi \in [0, \overline{\phi}],\ c \in [0, \tfrac{1}{4}],\ \frac{1}{n} H_{\phi,c}(h, f^*) \leq \frac{1}{n} \ln\left( \frac{16|\mathcal{F}|\overline{\phi}n^2}{\delta} \right) \right\}$$

$$A_2 := \left\{ \forall f \in \mathcal{F},\ \phi \in [0, \overline{\phi}],\ c \in [0, \tfrac{1}{4}], \right.$$

$$\left. -\frac{1}{n} H_{\phi,c}(f^*, f) \leq \mathbb{E}_x \left[ -\frac{\Delta_x \overline{\Delta}_{f^*,x,\phi,c}}{1 + \Delta_x \overline{\Delta}_{f^*,x,\phi,c}} + \frac{4}{3} \cdot \sigma_x^2 \overline{\Delta}_{f^*,x,\phi,c}^2 \right] + \frac{1}{n} \ln\left( \frac{48|\mathcal{F}|\overline{\phi}n^2}{\delta} \right) \right\}$$

$$A := A_1 \cap A_2$$

By Lemma 11, $\mathbb{P}(A_1) \geq 1 - \frac{\delta}{2}$; by Lemma 12, $\mathbb{P}(A_2) \geq 1 - \frac{\delta}{2}$. Taking a union bound, one can see that

$$\mathbb{P}(A) \geq 1 - \delta.$$

The subsequent reasoning conditions on $A$. $\forall f \in \mathcal{F}$, we have

$$L_n(f^*) - L_n(f)$$

$$= \max_{h \in \mathcal{F}, \phi' \in [0,\overline{\phi}], c' \in [0,\frac{1}{4}]} \min_{h' \in \mathcal{F}, \phi \in [0,\overline{\phi}], c \in [0,\frac{1}{4}]} \frac{1}{n} H_{\phi',c'}(h, f^*) - \frac{1}{n} H_{\phi,c}(h', f) \qquad \text{(definition of } L\text{)}$$

$$\leq \max_{h \in \mathcal{F}, \phi' \in [0,\overline{\phi}], c' \in [0,\frac{1}{4}]} \min_{\phi \in [0,\overline{\phi}], c \in [0,\frac{1}{4}]} \frac{1}{n} H_{\phi',c'}(h, f^*) - \frac{1}{n} H_{\phi,c}(f^*, f) \qquad (f^* \in \mathcal{F})$$

$$\leq \max_{h \in \mathcal{F}, \phi' \in [0,\overline{\phi}], c' \in [0,\frac{1}{4}]} \min_{\phi \in [0,\overline{\phi}], c \in [0,\frac{1}{4}]} \frac{1}{n} \ln\left( \frac{16|\mathcal{F}|\overline{\phi}n^2}{\delta} \right)$$

$$+ \mathbb{E}_x \left[ -\frac{\Delta_x \overline{\Delta}_{f^*,x,\phi,c}}{1 + \Delta_x \overline{\Delta}_{f^*,x,\phi,c}} + \frac{4}{3} \cdot \sigma_x^2 \overline{\Delta}_{f^*,x,\phi,c}^2 \right] + \frac{1}{n} \ln\left( \frac{48|\mathcal{F}|\overline{\phi}n^2}{\delta} \right)$$

$$\text{(definition of } A_1, A_2\text{)}$$

$$= \min_{\phi \in [0,\overline{\phi}], c \in [0,\frac{1}{4}]} \frac{1}{n} \ln\left( \frac{16|\mathcal{F}|\overline{\phi}n^2}{\delta} \right) + \mathbb{E}_x \left[ -\frac{\Delta_x \overline{\Delta}_{f^*,x,\phi,c}}{1 + \Delta_x \overline{\Delta}_{f^*,x,\phi,c}} + \frac{4}{3} \cdot \sigma_x^2 \overline{\Delta}_{f^*,x,\phi,c}^2 \right] + \frac{1}{n} \ln\left( \frac{48|\mathcal{F}|\overline{\phi}n^2}{\delta} \right)$$

$$\leq \min_{\phi \in [0,\overline{\phi}], c \in [0,\frac{1}{4}]} \mathbb{E}_x \left[ -\frac{\Delta_x \overline{\Delta}_{f^*,x,\phi,c}}{1 + \Delta_x \overline{\Delta}_{f^*,x,\phi,c}} + \frac{4}{3} \cdot \sigma_x^2 \overline{\Delta}_{f^*,x,\phi,c}^2 \right] + \frac{2}{n} \ln\left( \frac{48|\mathcal{F}|\overline{\phi}n^2}{\delta} \right)$$

That is,

$$\max_{\phi \in [0,\overline{\phi}]} \max_{c \in [0,\frac{1}{4}]} \underbrace{\mathbb{E}_x \left[ \frac{\Delta_x \overline{\Delta}_{f^*,x,\phi,c}}{1 + \Delta_x \overline{\Delta}_{f^*,x,\phi,c}} - \frac{4}{3} \cdot \sigma_x^2 \overline{\Delta}_{f^*,x,\phi,c}^2 \right]}_{=: \text{LHS}} \leq \frac{2}{n} \ln\left( \frac{48|\mathcal{F}|\overline{\phi}n^2}{\delta} \right) + (L_n(f) - L_n(f^*))$$

Recall that $\Delta_x = f_x^* - f_x$ and $\overline{\Delta}_{f^*,x,\phi,c} = \overline{(\phi(f_x^* - f_x))}_{[-c,c]}$. By Lemma 9, $\Delta_x \overline{\Delta}_{f^*,x,\phi,c} \geq 0$ for all $x$ and $U_x \leq \frac{1}{4}$.

Therefore,

$$\text{LHS} = \mathbb{E}_x \left[ \frac{\Delta_x \overline{\Delta}_{f^*,x,\phi,c}}{1 + \Delta_x \overline{\Delta}_{f^*,x,\phi,c}} - \frac{4}{3} \cdot \sigma_x^2 \overline{\Delta}_{f^*,x,\phi,c}^2 \right]$$

$$\geq \mathbb{E}_x \left[ \frac{4}{5} \Delta_x \overline{\Delta}_{f^*,x,\phi,c} - \frac{4}{3} \cdot \sigma_x^2 \overline{\Delta}_{f^*,x,\phi,c}^2 \right] \quad (\Delta_x \overline{\Delta}_{f^*,x,\phi,c} \geq 0, \ |\overline{\Delta}_{f^*,x,\phi,c}| \leq \frac{1}{4}, |\Delta_x| \leq 1)$$

$$= \mathbb{E}_x \left[ \frac{4}{5} |\Delta_x| \left( \phi |f^* - f| \wedge c \right) - \frac{4}{3} \cdot \sigma_x^2 \left( \phi |f^* - f| \wedge c \right)^2 \right]$$
$$(\overline{\Delta}_{f^*,x,\phi,c} = \text{sign}(f^* - f) \left( \phi |f^* - f| \wedge c \right))$$

$$= \frac{4}{5} \mathbb{E}_x \left[ |\Delta_x|^2 \left( \phi \wedge \frac{c}{|\Delta_x|} \right) \left[ 1 - \frac{5}{3} \cdot \sigma_x^2 \left( \phi \wedge \frac{c}{|\Delta_x|} \right) \right] \right]$$

We want to set $c$ and $\phi$ such that

$$\mathbb{E} \frac{1}{2} |\Delta_x|^2 \left( \phi \wedge \frac{c}{|\Delta_x|} \right) \geq \mathbb{E} \frac{5}{3} \cdot |\Delta_x|^2 \sigma_x^2 \left( \phi \wedge \frac{c}{|\Delta_x|} \right)^2, \tag{8}$$

which will give us the inequality of

$$\frac{4}{5} \mathbb{E} \frac{1}{2} |\Delta_x|^2 \left( \phi \wedge \frac{c}{|\Delta_x|} \right) \leq \frac{2}{n} \ln \left( \frac{48 |\mathcal{F}| \overline{\phi} n^2}{\delta} \right) + (L_n(f) - L_n(f^*)). \tag{9}$$

We choose $\phi$ such that $\phi = \frac{c}{\Delta^*}$ for some $\Delta^*$ to be chosen later, we can see that $\phi \wedge \frac{c}{|\Delta_x|} = c \left( \frac{1}{\Delta^*} \wedge \frac{1}{|\Delta_x|} \right)$. Using this, the above inequality (8) becomes:

$$\mathbb{E} \frac{1}{2} |\Delta_x|^2 c \left( \frac{1}{\Delta^*} \wedge \frac{1}{|\Delta_x|} \right) \geq \mathbb{E} \frac{5}{3} \cdot |\Delta_x|^2 c^2 \sigma_x^2 \left( \frac{1}{\Delta^*} \wedge \frac{1}{|\Delta_x|} \right)^2$$

We choose $c := c_0 \wedge \frac{1}{4}$, where

$$c_0 = \frac{\mathbb{E} \frac{1}{2} |\Delta_x|^2 \left( \frac{1}{\Delta^*} \wedge \frac{1}{|\Delta_x|} \right)}{\mathbb{E} \frac{5}{3} \cdot |\Delta_x|^2 \sigma_x^2 \left( \frac{1}{\Delta^*} \wedge \frac{1}{|\Delta_x|} \right)^2}$$

- If $c_0 \leq \frac{1}{4}$, then $c = c_0$. Plugging this into (9) along with the fact $\phi \wedge \frac{c}{|\Delta_x|} = c \left( \frac{1}{\Delta^*} \wedge \frac{1}{|\Delta_x|} \right)$, we have

$$\frac{4}{5} \left[ \mathbb{E} \frac{1}{2} |\Delta_x|^2 c \left( \frac{1}{\Delta^*} \wedge \frac{1}{|\Delta_x|} \right) \right]^2$$

$$\leq \mathbb{E} \frac{5}{3} \cdot |\Delta_x|^2 \sigma_x^2 \left( \frac{1}{\Delta^*} \wedge \frac{1}{|\Delta_x|} \right)^2 \cdot \left( \frac{2}{n} \ln \left( \frac{48 |\mathcal{F}| \overline{\phi} n^2}{\delta} \right) + (L_n(f) - L_n(f^*)) \right)$$

$$\leq \mathbb{E} \frac{5}{3} \sigma_x^2 \cdot \left( \frac{2}{n} \ln \left( \frac{48 |\mathcal{F}| \overline{\phi} n^2}{\delta} \right) + (L_n(f) - L_n(f^*)) \right)$$

$$\implies \left[ \mathbb{E} |\Delta_x|^2 \left( \frac{1}{\Delta^*} \wedge \frac{1}{|\Delta_x|} \right) \right]^2$$

$$\leq \frac{25}{12} \mathbb{E} \sigma_x^2 \cdot \left( \frac{2}{n} \ln \left( \frac{48 |\mathcal{F}| \overline{\phi} n^2}{\delta} \right) + (L_n(f) - L_n(f^*)) \right)$$

We could lower bound the LHS above by picking out the region with $|\Delta_x| \geq \Delta^*$ to arrive at:

$$\mathbb{E} \mathbb{1} \{ |\Delta_x| \geq \Delta^* \} |\Delta_x| \leq \sqrt{ \frac{25}{12} \mathbb{E} \sigma_x^2 \cdot \left( \frac{2}{n} \ln \left( \frac{48 |\mathcal{F}| \overline{\phi} n^2}{\delta} \right) + (L_n(f) - L_n(f^*)) \right) }$$

- If $c_0 > \frac{1}{4}$, then $c = \frac{1}{4}$. $c = \frac{1}{4} < c_0$ implies that (8) is true.

  Plugging $c = \frac{1}{4}$ into (9) along with the fact $\phi \wedge \frac{c}{|\Delta_x|} = c \left( \frac{1}{\Delta^*} \wedge \frac{1}{|\Delta_x|} \right)$, we have

  $$\frac{2}{5} \mathbb{E} |\Delta_x|^2 \left( \frac{1}{\Delta^*} \wedge \frac{1}{|\Delta_x|} \right) \leq \frac{2}{n} \ln \left( \frac{48|\mathcal{F}|\overline{\phi}n^2}{\delta} \right) + (L_n(f) - L_n(f^*))$$

  We could lower bound the LHS above by picking out the region with $|\Delta_x| \geq \Delta^*$ to arrive at:

  $$\frac{2}{5} \mathbb{E} \mathbb{1} \left\{ |\Delta_x| \geq \Delta^* \right\} |\Delta_x| \leq \frac{2}{n} \ln \left( \frac{48|\mathcal{F}|\overline{\phi}n^2}{\delta} \right) + (L_n(f) - L_n(f^*))$$

  $$\implies \mathbb{E} \mathbb{1} \left\{ |\Delta_x| \geq \Delta^* \right\} |\Delta_x| \leq 5\frac{1}{n} \ln \left( \frac{48|\mathcal{F}|\overline{\phi}n^2}{\delta} \right) + \frac{5}{2}(L_n(f) - L_n(f^*))$$

In either case, we have:

$$\mathbb{E} \mathbb{1} \left\{ |\Delta_x| \geq \Delta^* \right\} |\Delta_x| \leq \sqrt{\frac{25}{12} \mathbb{E} \sigma_x^2 \cdot \left( \frac{2}{n} \ln \left( \frac{48|\mathcal{F}|\overline{\phi}n^2}{\delta} \right) + (L_n(f) - L_n(f^*)) \right)}$$

$$+ \frac{5}{n} \ln \left( \frac{48|\mathcal{F}|\overline{\phi}n^2}{\delta} \right) + \frac{5}{2}(L_n(f) - L_n(f^*))$$

We choose $\Delta^* = \frac{1}{n} \ln \left( \frac{48|\mathcal{F}|\overline{\phi}n^2}{\delta} \right)$, which gives us,

$$\mathbb{E} \mathbb{1} \left\{ |\Delta_x| < \Delta^* \right\} |\Delta_x| \leq \frac{1}{n} \ln \left( \frac{48|\mathcal{F}|\overline{\phi}n^2}{\delta} \right)$$

Altogether, we have,

$$\mathbb{E}_x |\Delta_x|$$

$$\leq \sqrt{\frac{25}{12} \mathbb{E} \sigma_x^2 \cdot \left( \frac{2}{n} \ln \left( \frac{48|\mathcal{F}|\overline{\phi}n^2}{\delta} \right) + (L_n(f) - L_n(f^*)) \right)} + \frac{6}{n} \ln \left( \frac{48|\mathcal{F}|\overline{\phi}n^2}{\delta} \right) + \frac{5}{2}(L_n(f) - L_n(f^*))$$

We verify the choice $\phi$ is valid as follows.

$$\phi = \frac{c}{\Delta^*} \leq \frac{1}{4\Delta^*} \qquad\qquad (c \leq \tfrac{1}{4})$$

$$= \frac{1}{4\frac{1}{n} \ln \left( \frac{48|\mathcal{F}|\overline{\phi}n^2}{\delta} \right)}$$

$$= \frac{1}{4\frac{1}{n} \ln \left( \frac{12|\mathcal{F}|n^3}{\delta} \right)} \qquad\qquad (\overline{\phi} = \tfrac{n}{4})$$

$$\leq \overline{\phi}$$

which validates that $\phi \in [0, \overline{\phi}]$.

$\square$

## C   PROOF OF THEOREM 5

**Lemma 15.** *Recall that*

$$L_n(f) := \max_{h \in \mathcal{F}} \max_{\phi \in [0,\overline{\phi}]} \max_{c \in [0,\frac{1}{4}]} \frac{1}{n} \sum_{(x,y) \in D_n} \ln \left( 1 + (y - f_x)\overline{(\phi(h_x - f_x))}_{[-c,c]} \right)$$

*$L$ is $\frac{4}{3}n$-Lipchitz w.r.t. $\|\cdot\|_\infty$.*

*Proof.* For fixed $(h, \phi, c)$, define:

$$\Phi(f, h, \phi, c) := \frac{1}{n} \sum_{(x,y) \in D_n} \ln\left(1 + (y - f_x)\overline{\left(\phi(h_x - f_x)\right)}_{[-c,c]}\right).$$

We first show that $\Phi(f, h, \phi, c)$ is Lipschitz in $f$ w.r.t. $\|\cdot\|_\infty$.

Let $\varphi_1(t) := (y - t)\overline{\left(\phi(h_x - t)\right)}_{[-c,c]}$ for $t \in [0, 1]$ and $\varphi_2(t) := \ln(1 + t)$ for $t \in [-1/4, 1/4]$. If $\phi(h_x - t) \in [-c, c]$, then $|\varphi_1'(t)| = \phi|(t - y) + (t - h_x)| \leq \phi + c \leq n$; else if $\phi(h_x - t) \notin [-c, c]$, then $|\varphi_1'(t)| = c \leq \frac{1}{4}$. Hence $\varphi_1$ is $n$-Lipschitz. $|\varphi_2'(t)| = \frac{1}{1+t} \leq \frac{4}{3}$. Therefore, for any $(h, \phi, c)$, $\Phi(f, h, \phi, c)$ is $\frac{4}{3}n$-Lipschitz in $f$ w.r.t. $\|\cdot\|_\infty$:

$$\forall f, f' \in \mathcal{F}, \quad |\Phi(f, h, \phi, c) - \Phi(f', h, \phi, c)| \leq \frac{4}{3}n \cdot \|f - f'\|_\infty.$$

Furthermore, $\forall f, f' \in \mathcal{F}$,

$$L_n(f) - L_n(f') = \max_{h \in \mathcal{F}, \phi \in [0,\overline{\phi}], c \in [0,\frac{1}{4}]} \Phi(f, h, \phi, c) - \max_{h \in \mathcal{F}, \phi \in [0,\overline{\phi}], c \in [0,\frac{1}{4}]} \Phi(f', h, \phi, c)$$

$$\leq \max_{h \in \mathcal{F}, \phi \in [0,\overline{\phi}], c \in [0,\frac{1}{4}]} \Phi(f, h, \phi, c) - \Phi(f', h, \phi, c)$$

$$\leq \frac{4}{3}n \cdot \|f - f'\|_\infty$$

By symmetry,

$$L_n(f') - L_n(f) \leq \frac{4}{3}n \cdot \|f - f'\|_\infty.$$

Therefore, $\forall f, f' \in \mathcal{F}$,

$$|L_n(f) - L_n(f')| \leq \frac{4}{3}n \cdot \|f - f'\|_\infty.$$

$\square$

**Theorem 16** (Parametric class. Restatement of Theorem 5). *Assume the covering number of $\mathcal{F}$ satisfies Eqn. (3). Then, with probability at least $1 - \delta$, the output $\hat{f}$ of Algorithm 1 satisfies:*

$$\mathbb{E}_x |\hat{f}_x - f_x^*| \leq \sqrt{\frac{25}{3} \mathbb{E}\, \sigma_x^2 \frac{v}{n} \ln(\frac{12(1 + A)n^5}{\delta})} + 12\frac{v}{n}\ln(\frac{12(1 + A)n^5}{\delta})$$

*Proof.* Let $\mathcal{F}_\varepsilon$ be a minimum-cardinality $\varepsilon$-cover of $\mathcal{F}$ w.r.t. the metric $\|\cdot\|_\infty$. Then, we can designate $\hat{f}^\varepsilon \in \mathcal{F}_\varepsilon$ such that $\left\|\hat{f}^\varepsilon - \hat{f}\right\|_\infty \leq \varepsilon$.

Applying Theorem 14 with $\mathcal{F}' \leftarrow \mathcal{F}_\varepsilon \cup \{f^*\}$, we have

$$\mathbb{E}_x |\hat{f}_x^\varepsilon - f_x^*| \leq \sqrt{\frac{25}{12} \mathbb{E}\, \sigma_x^2 \cdot \left(2\frac{L}{n} + (L_n^\varepsilon(\hat{f}^\varepsilon) - L_n^\varepsilon(f^*))\right)} + 6\frac{L}{n} + \frac{5}{2}(L_n^\varepsilon(\hat{f}^\varepsilon) - L_n^\varepsilon(f^*)),$$

where

$$L = \ln\left(\frac{48|\mathcal{F}'|\overline{\phi}n^2}{\delta}\right)$$

$$L_n^\varepsilon(f) := \max_{h \in \mathcal{F}'} \max_{\phi \in [0,\overline{\phi}]} \max_{c \in [0,\frac{1}{4}]} \frac{1}{n} \sum_{(x,y) \in D_n} \ln\left(1 + (y - f_x)\overline{\left(\phi(h_x - f_x)\right)}_{[-c,c]}\right)$$

For analysis purposes, we also denote:

$$L_n(f) := \max_{h \in \mathcal{F}} \max_{\phi \in [0,\overline{\phi}]} \max_{c \in [0,\frac{1}{4}]} \frac{1}{n} \sum_{(x,y) \in D_n} \ln\left(1 + (y - f_x)\overline{\left(\phi(h_x - f_x)\right)}_{[-c,c]}\right)$$

Recall that our Algorithm 1 returns $\hat{f} \in \arg\min_{f \in \mathcal{F}} L_n(f)$. We have,

$$L_n^\varepsilon(\hat{f}^\varepsilon) - L_n^\varepsilon(f^*) = \left(L_n^\varepsilon(\hat{f}^\varepsilon) - L_n^\varepsilon(\hat{f})\right) + \left(L_n^\varepsilon(\hat{f}) - L_n(\hat{f})\right) + \left(L_n(\hat{f}) - L_n(f^*)\right) + \left(L_n(f^*) - L_n^\varepsilon(f^*)\right)$$

$$\leq \frac{4}{3}n \cdot \left\|\hat{f}^\varepsilon - \hat{f}\right\|_\infty + 0 + 0 + \left(L_n(f^*) - L_n^\varepsilon(f^*)\right) \tag{10}$$

where the first term is by Lemma 15, the second term is by the definition of $L_n^\varepsilon$ and $L_n$, and the third term is by the definition of $\hat{f}$.

We bound $L_n(f^*) - L_n^\varepsilon(f^*)$ as follows. Let $h^*$ be such that

$$L_n(f^*) = \max_{\phi \in [0,\overline{\phi}]} \max_{c \in [0,\frac{1}{4}]} \frac{1}{n} \sum_{(x,y) \in D_n} \ln\left(1 + (y - f_x^*)\overline{\left(\phi(h_x^* - f_x^*)\right)}_{[-c,c]}\right).$$

Let $h_\varepsilon^* \in \mathcal{F}_\varepsilon$ be such that $\|h^* - h_\varepsilon^*\| \leq \varepsilon$. Then,

$$L_n(f^*) - L_n^\varepsilon(f^*)$$

$$\leq \max_{\phi \in [0,\overline{\phi}]} \max_{c \in [0,\frac{1}{4}]} \frac{1}{n} \sum_{(x,y) \in D_n} \ln\left(1 + (y - f_x^*)\overline{\left(\phi(h_x^* - f_x^*)\right)}_{[-c,c]}\right)$$

$$- \max_{\phi \in [0,\overline{\phi}]} \max_{c \in [0,\frac{1}{4}]} \frac{1}{n} \sum_{(x,y) \in D_n} \ln\left(1 + (y - f_x^*)\overline{\left(\phi(h_{\varepsilon,x}^* - f_x^*)\right)}_{[-c,c]}\right)$$

Note that for any $(\phi, c)$,

$$\left|(y - f^*)\overline{\left(\phi \cdot (h^* - f^*)\right)}_{[-c,c]} - (y - f^*)\overline{\left(\phi \cdot (h_\varepsilon^* - f^*)\right)}_{[-c,c]}\right|$$

$$= |(y - f^*)| \cdot \left|\overline{\left(\phi \cdot (h^* - f^*)\right)}_{[-c,c]} - \overline{\left(\phi \cdot (h_\varepsilon^* - f^*)\right)}_{[-c,c]}\right|$$

$$\leq |(y - f^*)| \cdot \frac{n}{4}\varepsilon \qquad\qquad (\phi \in [0, \tfrac{n}{4}])$$

Using $t \mapsto \ln(1 + t)$ is $\frac{4}{3}$-Lipschitz for $t \in [-\frac{1}{4}, \frac{1}{4}]$,

$$L_n(f^*) - L_n^\varepsilon(f^*) \leq \frac{4}{3} \cdot \frac{n}{4}\varepsilon \cdot \frac{1}{n}\sum_{(x,y)} |(y - f_x^*)|$$

$$\leq \frac{1}{3}n\varepsilon.$$

Plugging back into Eqn. (10),

$$L_n^\varepsilon(\hat{f}^\varepsilon) - L_n^\varepsilon(f^*) \leq \frac{4}{3}n \cdot \left\|\hat{f}^\varepsilon - \hat{f}\right\|_\infty + \left(L_n(f^*) - L_n^\varepsilon(f^*)\right) \leq 2n\varepsilon \tag{11}$$

Therefore,

$$\mathbb{E}_x |\hat{f}_x - f_x^*|$$

$$\leq \mathbb{E}_x |\hat{f}_x - \hat{f}_x^\varepsilon| + \mathbb{E}_x |\hat{f}_x^\varepsilon - f_x^*| \qquad\qquad \text{(Triangle inequality)}$$

$$\leq \varepsilon + \sqrt{\frac{25}{12} \mathbb{E}\,\sigma_x^2 \cdot \left(2\frac{L}{n} + (L_n^\varepsilon(\hat{f}^\varepsilon) - L_n^\varepsilon(f^*))\right)} + 6\frac{L}{n} + \frac{5}{2}(L_n^\varepsilon(\hat{f}^\varepsilon) - L_n^\varepsilon(f^*)) \quad \text{(Theorem 14)}$$

$$\leq \varepsilon + \sqrt{\frac{25}{12} \mathbb{E}\,\sigma_x^2 \cdot \left(2\frac{L}{n} + 2n\varepsilon\right)} + 6\frac{L}{n} + \frac{5}{2}(2n\varepsilon) \qquad\qquad \text{(Eqn. (11))}$$

$$\leq \sqrt{\frac{25}{12} \mathbb{E}\,\sigma_x^2\left(\frac{2}{n}\ln(\frac{(1 + (A/\varepsilon)^v)48\overline{\phi}n^2}{\delta}) + 2n\varepsilon\right)} + 6\frac{1}{n}\ln(\frac{(1 + (A/\varepsilon)^v)48\overline{\phi}n^2}{\delta}) + 6n\varepsilon$$

where the last inequality is because the covering number of $\mathcal{F}$ satisfies Eqn. (3), i.e., for every $\varepsilon > 0$, $N(\varepsilon, \mathcal{F}, \|\cdot\|_\infty) \leq (\frac{A}{\varepsilon})^v$.

Choosing $\varepsilon = \frac{1}{n^2}$ gives:

$$\mathbb{E}_x |\hat{f}_x - f_x^*| \leq \sqrt{\frac{25}{3} \mathbb{E}\,\sigma_x^2 \frac{v}{n}\ln(\frac{12(1 + A)n^5}{\delta})} + 12\frac{v}{n}\ln(\frac{12(1 + A)n^5}{\delta})$$

$$\square$$

# D  PROOF OF COROLLARY 6

**Corollary 17** (Linear class. Restatement of Corollary 6). *Let $\mathcal{F}$ be a linear function class in $d$-dimensional space: $\mathcal{F} = \{x \mapsto x^\top \theta + \frac{1}{2} : \|\theta\|_2 \leq \frac{1}{2}\}$ and the instance space $\mathcal{X} = \{x \in \mathbb{R}^d : \|x\|_2 \leq 1\}$. Then, with probability at least $1 - \delta$, the output $\hat{f}$ of Algorithm 1 satisfies:*

$$\mathbb{E}_x |\hat{f}_x - f_x^*| \leq \sqrt{\frac{25}{3} \mathbb{E} \sigma_x^2 \frac{d}{n} \ln(\frac{36n^5}{\delta})} + 12\frac{d}{n} \ln(\frac{36n^5}{\delta})$$

*Proof.* To make use of Theorem 5, we just need to show that the $L_\infty$ covering number of this class, $N(\varepsilon, \mathcal{F}, \|\cdot\|_\infty)$, grows polynomially in $1/\varepsilon$. The covering number of the linear class is not new (e.g., Exercise 20.3 of Lattimore and Szepesvári (2018)), we include the proof for completeness.

Denote by $\mathcal{W}$ the parameter space: $\mathcal{W} := \{\theta : \|\theta\|_2 \leq \frac{1}{2}\}$.

Let $f_u$ and $f_v$ be two functions in $\mathcal{F}$. We have,

$$\begin{aligned}
\|f_u - f_v\|_\infty &= \sup_{x \in \mathcal{X}} |x^\top u - x^\top v| \\
&= \sup_{x \in \mathcal{X}} |x^\top (u - v)| \\
&\leq \sup_{x \in \mathcal{X}} \|u - v\|_2 \|x\|_2 \qquad \text{(Cauchy-Schwarz)} \\
&\leq \|u - v\|_2 \qquad (\|x\|_2 \leq 1)
\end{aligned}$$

This implies that an $\varepsilon$-cover of the parameter space $\mathcal{W}$ induces an $\varepsilon$-cover of the function class $\mathcal{F}$. Therefore, we can bound the covering number of the function class by the covering number of the parameter space:

$$N(\varepsilon, \mathcal{F}, \|\cdot\|_\infty) \leq N(\varepsilon, \mathcal{W}, \|\cdot\|_2) \tag{12}$$

The problem is now reduced to finding the covering number of the parameter space $\mathcal{W}$, which is a ball of radius $\frac{1}{2}$ in a $d$-dimensional Euclidean space. This is a standard geometric result. The number of $\varepsilon$-balls to cover a ball of radius $B$ is bounded by:

$$N(\varepsilon, \mathcal{W}, \|\cdot\|_2) \leq \left(\frac{2B}{\varepsilon} + 1\right)^d = \left(\frac{1}{\varepsilon} + 1\right)^d \leq \left(\frac{2}{\varepsilon}\right)^d \text{ for } \varepsilon \leq \frac{1}{2}$$

Combining with Eqn. (12), we arrive at:

$$N(\varepsilon, \mathcal{F}, \|\cdot\|_\infty) \leq \left(\frac{2}{\varepsilon}\right)^d$$

Applying Theorem 5 with $v = d$ and $A = 2$ concludes the proof. $\qquad \square$

# E  ON THE CONJECTURE OF INCONSISTENCY

**Theorem 18.** *Consider the following problem instance. Let $\mathcal{X} = [0, 1]$, and $\mathcal{F} \subset \{\mathcal{X} \to [0, 1]\}$ be the Lipschitz function class with Lipschitz parameter $L = 101$, i.e., $\forall f \in \mathcal{F}, \forall x, x' \in \mathcal{X}, |f(x) - f(x')| \leq L \cdot |x - x'|$. Let $\mathcal{D}_X$, the marginal distribution of $X$, be the uniform distribution on $\mathcal{X}$, and $\forall x \in \mathcal{X}, \mathcal{D}_{Y|X=x}$ be the Bernoulli distribution with parameter $x$.*

*For this problem instance, the following holds:*

*there exists $f_0 \in \mathcal{F}, f_0 \neq f^*$, constants $N > 0$ and $c_0, c_1 > 0$, such that $\forall n \geq N$,*

$$\mathbb{P}(L_n(f^*) - L_n(f_0) > c_1) \geq c_0 .$$

*Proof.* In this instance, the true regression function is $f_x^* = \mathbb{E}[Y \mid X = x] = x$. Note that $f^*$ is 1-Lipschitz, so $f^* \in \mathcal{F}$.

We construct $f_0$ as follows:

$$f_0(x) = \begin{cases} 0 & x \in [0, 1/4] \\ 2x - 1/2 & x \in [1/4, 3/4] \\ 1 & x \in [3/4, 1] \end{cases}$$

Recall that $\forall f \in \mathcal{F}$,

$$H_{\phi,c}(h, f) := \sum_{(x,y) \in D_n} \ln\left(1 + (y - f_x)\overline{(\phi(h_x - f_x))}_{[-c,c]}\right)$$

$$L_n(f) := \max_{h \in \mathcal{F}} \max_{\phi \in [0, \bar{\phi}]} \max_{c \in [0, \frac{1}{4}]} \frac{1}{n} H_{\phi,c}(h, f)$$

We may overload the notation $L_n(f)$ to $L_n(f, h, \phi, c)$, to denote the dependence on $h, \phi, c$.

Note that on any datapoint $(x, y)$,

$$\max_{h,\phi,c}(y - f_{0,x})\overline{(\phi(h_x - f_{0,x}))}_{[-c,c]} \leq |y - f_{0,x}| \cdot c \leq |y - f_{0,x}| \cdot \frac{1}{4}$$

Thus,

$$L_n(f_0) \leq \frac{1}{n} \sum_{(x,y) \in D_n} \max_{h,\phi,c} \ln\left(1 + (y - f_{0,x})\overline{(\phi(h_x - f_{0,x}))}_{[-c,c]}\right)$$

$$\leq \frac{1}{n} \sum_{(x,y) \in D_n} \ln\left(1 + |y - f_{0,x}| \cdot \frac{1}{4}\right) := \ell_{f_0}$$

As $n \to \infty$, by the weak law of large numbers, the sample average converges in probability to its true expectation:

$$\ell_{f_0} = \frac{1}{n} \sum_{(x,y) \in D_n} \ln\left(1 + |y - f_{0,x}| \cdot \frac{1}{4}\right) \xrightarrow{p} \mathbb{E}\ln\left(1 + |y - f_{0,x}| \cdot \frac{1}{4}\right) = 0.0625$$

By the definition of convergence in probability, for every $\varepsilon > 0$,

$$\lim_{n \to \infty} \mathbb{P}(|\ell_{f_0} - 0.0625| < \varepsilon) = 1,$$

which implies that there exists $N_1$, such that $\forall n \geq N_1$,

$$\mathbb{P}(|\ell_{f_0} - 0.0625| < 0.0005) \geq 0.9$$
$$\implies \mathbb{P}(\ell_{f_0} < 0.0630) \geq 0.9$$
$$\implies \mathbb{P}(L_n(f_0) < 0.0630) \geq 0.9 \tag{13}$$

Next, we turn to lower bounding $L_n(f^*)$ for large enough $n$.

Recall that in our problem, $X$ follows a uniform distribution on $[0, 1]$. In a sample drawn from this distribution, let a "gap" be the distance between two adjacent samples, i.e., $X_{(j)} - X_{(j-1)}$ for any $j \in [1, n]$ as well as $X_{(1)}$, where $X_{(j)}$ is the increasingly sorted sample.

We call it a "good" gap if it is $\geq \frac{0.0202}{n}$. By Lemma 19, if $n \geq 10^4$, then with probability at least 0.9, the number of "good" gaps is at least $0.97n$. Suppose the event that "the number of 'good' gaps is at least $0.97n$" happens.

We choose $(h^*, \phi^*, c^*)$ (not necessarily a maximizer) such that:

- $h_x^* - f_x^* = \text{sign}(y - f_x^*) \cdot \frac{1}{4}$ for only those data points next to a good gap. Indeed, such $h^*$ is ($L = 101$)-Lipschitz: let $x_{(j)}$ be an observed datapoint next to a good gap, $x_{(j-)}$ be the immediate previous datapoint that is next to a good gap, then $x_{(j)} - x_{(j-)} \geq \frac{0.0202}{n}$, hence,

$$\left|h_{x_{(j)}}^* - h_{x_{(j-)}}^*\right| = \left|(h_{x_{(j)}}^* - f_{x_{(j)}}^*) - (h_{x_{(j-)}}^* - f_{x_{(j-)}}^*) + f_{x_{(j)}}^* - f_{x_{(j-)}}^*\right|$$

$$= \left| \text{sign}(y_{(j)} - f^*_{x_{(j)}}) \cdot \frac{1}{n} - \text{sign}(y_{(j^-)} - f^*_{x_{(j^-)}}) \cdot \frac{1}{n} + x_{(j)} - x_{(j^-)} \right|$$
$$(f^*_x = x)$$

$$\leq \frac{2}{n} + (x_{(j)} - x_{(j^-)}) \qquad \text{(Triangle inequality; } x_{(j)} - x_{(j^-)} \geq 0)$$

$$\leq (\frac{2}{0.0202} + 1)(x_{(j)} - x_{(j^-)}) \qquad (x_{(j)} - x_{(j^-)} \geq \frac{0.0202}{n})$$

$$\leq 101 \cdot (x_{(j)} - x_{(j^-)})$$

which means such $h^*$ exists in $\mathcal{F}$.

- $c^* = \frac{1}{4}$.

- $\phi^* = \frac{n}{4}$.

Thus, $L_n(f^*) \geq L_n(f^*, h^*, \phi^*, c^*)$.

Let $z := |y - f^*_x|$ and sort $z_i$ in increasing order as $z_{(i)}$. We can see that

$$L_n(f^*, h^*, \phi^*, c^*) \geq \frac{1}{n} \left[ \sum_{i=1}^{0.97n} \ln\left(1 + z_{(i)} \cdot \frac{1}{4}\right) + \sum_{i=0.97n}^{n} \ln\left(1 - z_{(i)} \cdot \frac{1}{4}\right) \right] := \ell_{f^*} \qquad (14)$$

As $n \to \infty$, by the weak law of large numbers,

$$\ell_{f^*} = \frac{1}{n} \left[ \sum_{i=1}^{0.97n} \ln\left(1 + z_{(i)} \cdot \frac{1}{4}\right) + \sum_{i=0.97n}^{n} \ln\left(1 - z_{(i)} \cdot \frac{1}{4}\right) \right]$$

$$\xrightarrow{p} \int_0^{q_{0.97}} \ln\left(1 + z \cdot \frac{1}{4}\right) f_Z(z) \, \mathrm{d}z + \int_{q_{0.97}}^1 \ln\left(1 - z \cdot \frac{1}{4}\right) f_Z(z) \, \mathrm{d}z \qquad (15)$$

where $q_{0.97}$ is the 97th percentile of the distribution of $Z$.

We calculate the CDF and PDF of $Z$:

First note that

$$\mathbb{P}(Z \leq z \mid X = x) = \mathbb{P}(|Y - X| \leq z \mid X = x)$$
$$= (1 - x) \mathbb{1}\{x \leq z\} + x \mathbb{1}\{1 - x \leq z\}$$

The reason is: if $Z = x$ then $Z \leq z$ iff $x \leq z$; if $Z = 1 - x$ then $Z \leq z$ iff $1 - x \leq z$. Thus for $z \in [0, 1]$,

$$F_Z(z) = \mathbb{P}(Z \leq z)$$
$$= \mathbb{E}[\mathbb{P}(Z \leq z) \mid X] \qquad \text{(law of total probability)}$$

$$= \int_0^1 [(1 - x) \mathbb{1}\{x \leq z\} + x \mathbb{1}\{1 - x \leq z\}] \, \mathrm{d}x$$

$$= \int_0^1 [(1 - x) \mathbb{1}\{x \leq z\} + x \mathbb{1}\{1 - z \leq x\}] \, \mathrm{d}x$$

$$= \int_0^z (1 - x) \, \mathrm{d}x + \int_{1-z}^1 x \, \mathrm{d}x$$

$$= 2 \int_0^z (1 - x) \, \mathrm{d}x \qquad \text{(symmetry)}$$

$$= 2(z - \frac{z^2}{2})$$

$$= 2z - z^2$$

Hence,

$$F_Z(z) = 2z - z^2, \ z \in [0, 1]$$

$$f_Z(z) = 2(1-z), \ z \in [0,1]$$

Setting $F_Z(z) = 0.97$, the solution that lies within $[0,1]$ is:

$$q_{0.97} = 1 - \sqrt{1-0.97} \approx 0.8268$$

We have,

$$\int_0^{q_{0.97}} \ln\left(1 + z \cdot \frac{1}{4}\right) f_Z(z)\, \mathrm{d}z + \int_{q_{0.97}}^1 \ln\left(1 - z \cdot \frac{1}{4}\right) f_Z(z)\, \mathrm{d}z$$

$$= \int_0^{q_{0.97}} \ln\left(1 + z \cdot \frac{1}{4}\right) 2(1-z)\, \mathrm{d}z + \int_{q_{0.97}}^1 \ln\left(1 - z \cdot \frac{1}{4}\right) 2(1-z)\, \mathrm{d}z$$

$$= 0.0726 - 0.0075$$

$$= 0.0651 \tag{16}$$

Putting together the above calculations (Eqns. (15) to (16)), we have, as $n \to \infty$,

$$\ell_{f^*} = \frac{1}{n}\left[\sum_{i=1}^{0.97n} \ln\left(1 + z_{(i)} \cdot \frac{1}{4}\right) + \sum_{i=0.97n}^{n} \ln\left(1 - z_{(i)} \cdot \frac{1}{4}\right)\right] \xrightarrow{p} 0.0651$$

By the definition of convergence in probability, for every $\varepsilon > 0$,

$$\lim_{n \to \infty} \mathbb{P}(|\ell_{f^*} - 0.0651| < \varepsilon) = 1,$$

which implies that there exists $N_2$, such that $\forall n \geq N_2$,

$$\mathbb{P}(|\ell_{f^*} - 0.0651| < 0.0005) \geq 0.9$$

$$\implies \mathbb{P}(\ell_{f^*} > 0.0646) \geq 0.9 \tag{17}$$

Combining $L_n(f^*) \geq L_n(f^*, h^*, \phi^*, c^*)$, Eqns. (14) (17) and taking a union bound with the event "the number of 'good' gaps is at least $0.97n$", which, as we mentioned and by Lemma 19, happens with probability at least 0.9, we have, $\forall n \geq \max\left\{10^4, N_2\right\}$,

$$\mathbb{P}(L_n(f^*) > 0.0646) \geq 0.8 \tag{18}$$

Combining Equations (13) and (18) with a union bound, we get $\forall n \geq \max\left\{10^4, N_1, N_2\right\}$,

$$\mathbb{P}(L_n(f_0) < 0.0630 < 0.0646 < L_n(f^*)) \geq 0.7$$

$$\square$$

**Lemma 19.** *Suppose $n \geq 10^4$. In a sample set drawn i.i.d. from a uniform distribution on $[0,1]$, let a "gap" be the distance between two adjacent samples.i.e., $X_{(j)} - X_{(j-1)}$ for any $j \in [1,n]$ as well as $X_{(1)}$ and where $X_{(j)}$ is the increasingly sorted sample.*

*We call a cap "good" if it is $\geq \frac{k}{n}$, where $k = -\ln 0.98 = 0.0202$.*

*With probability at least 0.9, the number of "good" gaps is at least $0.97n$.*

*Proof.* In a sample drawn from the distribution on $[0,1]$, by symmetry, all gaps i.e., $X_{(j)} - X_{(j-1)}$ for any $j \in [1,n]$ as well as $X_{(1)}$ where $X_{(j)}$ is the sorted sample, have the same probability distribution.

To study the properties of these gaps, we will find the distribution of the first gap, $X_{(1)}$, which is the simplest to compute.

We compute the CDF:

$$\begin{aligned}
F_{X_{(1)}}(x) &= \mathbb{P}(X_{(1)} \leq x) \\
&= 1 - \mathbb{P}(X_{(1)} > x) \\
&= 1 - \mathbb{P}(\forall i \in [n], \ X_i > x) \\
&= 1 - (1-x)^n
\end{aligned}$$

Let the random variable $G$ be a gap, that is, $G \overset{\mathrm{d}}{=} X_{(1)}$.

Recall that from the CDF of $G$, we have $P(G \leq t) = 1 - (1-t)^n$. Hence $P(G \geq \frac{k}{n}) = (1 - \frac{k}{n})^n \xrightarrow{n \to \infty} e^{-k}$. Since $k = -\ln 0.98 = 0.0202$, we have w.p. 0.98, any single gap satisfies $G \geq \frac{0.0202}{n}$, i.e., w.p. 0.98, any single gap is good.

Let $I_i$ be an indicator random variable for the $i$-th gap, where $i \in [n]$. $I_i = 1$ if the $i$-th gap is "good", i.e., $G_i \geq \frac{k}{n}$ and $I_i = 0$ otherwise. Hence, $\mathbb{P}(I_i = 1) = p = 0.98$.

Let $Y$ be the total number of good gaps: $Y = \sum_{i=1}^n I_i$. By linearity of expectation,

$$\mathbb{E}[Y] = \mathbb{E}[\sum_{i=1}^n I_i] = 0.98n.$$

Our goal is to find an $L$ such that $\mathbb{P}(Y \geq L) \geq 0.9$.

Note that

$$\mathrm{Var}(Y) = \mathrm{Var}\left(\sum_{i=1}^n I_i\right) = \sum_{i=1}^n \mathrm{Var}(I_i) + \sum_{i \neq j} \mathrm{Cov}(I_i, I_j)$$

We argue that $\mathrm{Cov}(I_i, I_j) \leq 0$ for $i \neq j$. The intuitive interpretation is that, since the gaps are not independent, if one gap is very large, the others must be smaller to compensate, since their total length is fixed. This means they are negatively correlated, so $\mathrm{Cov}(I_i, I_j) \leq 0$ for $i \neq j$. More formally, one can verify (by exchangeability) that for $i \neq j$,

$$\mathrm{Cov}(I_i, I_j) = \mathbb{E}[I_i I_j] - \mathbb{E}[I_i]\,\mathbb{E}[I_j]$$

$$\mathbb{E}[I_i] = \mathbb{P}(I_i = 1) = (1 - \frac{k}{n})^n$$

$$\mathbb{E}[I_j] = \mathbb{P}(I_j = 1) = (1 - \frac{k}{n})^n$$

$$\mathbb{E}[I_i I_j] = \mathbb{P}(I_i = 1, I_i = j) = (1 - 2\frac{k}{n})^n$$

$$\implies \mathrm{Cov}(I_i, I_j) \leq 0$$

Thus,

$$\mathrm{Var}(Y) \leq \sum_{i=1}^n \mathrm{Var}(I_i) = np(1-p) = 0.0196n$$

Applying Chebyshev's inequality,

$$\mathbb{P}(Y \leq \mathbb{E}[Y] - \varepsilon) \leq \frac{\mathrm{Var}(Y)}{\varepsilon^2}$$

$$\implies \mathbb{P}(Y \leq 0.98n - \varepsilon) \leq \frac{np(1-p)}{\varepsilon^2} = \frac{0.0196n}{\varepsilon^2}$$

Solving $\frac{0.0196n}{\varepsilon^2} = 0.1$, we get $\varepsilon = \sqrt{0.196n}$. Therefore, with $n \geq 10^4$, with probability at least 0.9,

$$Y \geq 0.98n - \sqrt{0.196n} \geq 0.97n.$$

$\square$

**The implication:** Theorem 18 shows that, for sufficiently large samples, with constant probability, there exists a sufficiently large gap between $L_n(f^*)$ and $L_n(f_0)$.

This theorem may support our conjecture about the inconsistency of the betting loss ERM estimator for nonparametric classes. The idea is that, if an ERM estimator is consistent, the true regression function $f^*$ should effectively become the minimizer of the loss for large samples. However, Theorem 18 shows that this is demonstrably not true.

In more detail, let $L_n^{\min} = \min_{f \in \mathcal{F}} L_n(f)$. By definition, $L_n(\hat{f}) = L_n^{\min}$.

By Theorem 18, for all $n \geq N$,

$$\mathbb{P}(L_n(f^*) - L_n(f_0) > c_1) \geq c_0 \, .$$

Since $L_n^{\min} \leq L_n(f_0)$, this implies:

$$\mathbb{P}(L_n(f^*) - L_n^{\min} > c_1) \geq \mathbb{P}(L_n(f^*) - L_n(f_0) > c_1) \geq c_0 \, .$$

We conjecture that the significant difference of $L_n(f^*) - L_n(\hat{f})$ will effectively translate to a notable distance between $\hat{f}$ and $f^*$.

We interpret this negative result as a structural limitation: while betting loss adapts optimally in parametric regimes, it is misaligned with $L_2$ risk minimization under Lipschitz constraints. Intuitively, the multiple max operations that define the betting loss act as a "robustification" step: in sufficiently rich classes like Lipschitz, this mechanism alters the geometry of the optimization, making betting loss behave as if it were minimizing an $L_1$ criterion. As a result, the predictor is pulled toward the conditional median rather than the conditional mean $f^*$, leading to inconsistency in nonparametric settings.

## F    COMPARING THE TWO FIRST-ORDER QUANTITIES

In this section, we first show that $\mathbb{E}_x[f^*(x) \wedge (1 - f^*(x))] \leq \mathbb{E}_x[f^*(x)] \wedge \mathbb{E}_x[1 - f^*(x)]$, then we give an example where the difference between these two quantities can be arbitrarily large.

**Lemma 20.** *Recall that $f^* : \mathcal{X} \to [0,1]$. We have,*

$$\mathbb{E}_x[f^*(x) \wedge (1 - f^*(x))] \leq \mathbb{E}_x[f^*(x)] \wedge \mathbb{E}_x[1 - f^*(x)]$$

*Proof.* Note that

$$\mathbb{E}_x[f^*(x) \wedge (1 - f^*(x))] \leq \mathbb{E}_x[f^*(x)],$$

and

$$\mathbb{E}_x[f^*(x) \wedge (1 - f^*(x))] \leq \mathbb{E}_x[1 - f^*(x)].$$

Hence,

$$\mathbb{E}_x[f^*(x) \wedge (1 - f^*(x))] \leq \mathbb{E}_x[f^*(x)] \wedge \mathbb{E}_x[1 - f^*(x)].$$

$\square$

**Example 21.** *Let $\varepsilon > 0$ be a small number, $Y$ be of the distribution $\mathbb{P}(Y = \varepsilon) = \mathbb{P}(Y = 1 - \varepsilon) = \frac{1}{2}$. Then,*

$$\mathbb{E}[Y \wedge 1 - Y] = \varepsilon,$$

*whereas*

$$\mathbb{E}[Y] \wedge \mathbb{E}[1 - Y] = \frac{1}{2}.$$

## G    PROOF OF LEMMA 2

*Proof.*

$$\begin{aligned}
\mathrm{Var}(Y) &= \mathbb{E}[(Y - \mathbb{E}[Y])^2] \\
&= \mathbb{E}[Y^2 - 2Y\,\mathbb{E}[Y] + \mathbb{E}^2[Y]] \\
&\leq \mathbb{E}[Y - 2Y\,\mathbb{E}[Y] + \mathbb{E}^2[Y]] && (Y \in [0,1]) \\
&= \mathbb{E}[Y] - \mathbb{E}^2[Y] \\
&= \mathbb{E}[Y](1 - \mathbb{E}[Y])
\end{aligned}$$

One can see that the equality in the third line is attained iff $Y$ is Bernoulli distributed.    $\square$

