# OpenReview forum: "Second-Order Bounds for [0,1]-Valued Regression via Betting Loss"
_ICLR.cc/2026/Conference — Submitted to ICLR 2026_

### Official Review · Reviewer_YX2n · 2025-10-29

**Soundness:** 3
**Presentation:** 3
**Contribution:** 3
**Rating:** 8
**Confidence:** 3

**Summary:**

The paper presents a theoretical study on generalisation error bounds for [0, 1]-valued regression in the stochastic setting. The main contribution is a “second-order” bound for a proposed novel loss function. First, a first-order bound is shown for log-loss. This bound scales with a term that upperbounds the conditional variance. Next, a novel betting loss is proposed, which has two hyperparameters $\bar{\phi}$ and $c$. A second-order bound is shown for this loss, with the bound depending on the conditional variance $\sigma_x$. This result is initially shown for a finite hypothesis class and then extended to a broader family of functions defined as “parametric” class. Computational experiments have been performed on a synthetic dataset. Improved generalisation using betting loss has been shown in comparison to the squared and logit loss. The experiments were conducted with varying levels of conditional variance in the data.

**Strengths:**

*) A novel loss function, referred to as betting loss, is proposed for the regression problem.

*) The minimiser of this loss guarantees a second-order generalisation error bound. The second-orderness is implied by dependence on conditional variance. The minimiser can be attained without knowledge of the conditional variance

*) Second-order bounds have been established for the finite hypothesis class case and a subclass of the infinite hypothesis class, specifically the parametric class. A lack of such bounds for the non-parametric class is conjectured, and a supporting theory for it is detailed in the appendix.

*) Experimental validation of improved generalisation using betting loss is done using a synthetic dataset. Experiments are shown with varying levels of conditional variance. The experiments are repeated multiple times, and the standard errors of the performances are reported.

*)The related works have been discussed in an organised manner, with the distinction of the current work highlighted.

**Weaknesses:**

*) The betting loss function has two parameters \bar{\phi} and c. There is a lack of discussion on the effect of these parameters.

*) More discussion of this loss function within the broader class of loss functions will help readers.

*) The minimiser of the betting loss cannot be accurately computed because of $\max_h \max_\phi \max_c$. As done in the experiments, it requires performing a discretised search. This increases computations.

*) No experiments on a real-world dataset.

**Questions:**

*) Can a discussion be provided on the parameters $\phi$ and $c$?

*) Can some experiments be shown on a real-world dataset?

---

> ### Author Response · Authors · 2025-11-23
>
> We sincerely thank the reviewer for their positive assessment, and for the constructive feedback.
>
> **[Role of the parameters $\phi$ and $c$]**
>
> It is important to note that $\phi$ and $c$ are **not** user-facing hyperparameters that need to be tuned. Instead, they are internal variables that are part of the min-max optimization (Algorithm 1).
>
> * **Role of $\phi$ (Bet magnitude):** This parameter scales the "bet" ($h_x - f_x$). By maximizing over $\phi \in [0, \overline{\phi}]$ (where $\overline{\phi}=n/4$), the algorithm ensures that our final function $\hat{f}$ is robust even when an adversary $h$ can choose the most effective bet size to challenge it.
> * **Role of $c$ (Clipping):** This is a clipping parameter, also part of the internal maximization. Its primary role is theoretical (such as in the proof of Lemma 9): it ensures the stability of the loss function by keeping the argument of the logarithm not too far away from 1.
>
> **[Real-world dataset experiments]**
>
> For experiments on the real-world datasets suggested by the reviewer, we regenerated the labels to ensure realizability. All experiment settings are identical to those in the main paper. We experiment on the Wine Quality datasets (Cortez et al., 2009), where the feature vectors are normalized via Min-Max scaling. As shown in Table 1 and Table 2, the betting loss consistently achieves the lowest MAE. (Reported values indicate mean $\pm$ standard error. Best results are marked in **bold**.)
>
> **Table 1: Comparison of average mean absolute error (MAE) on the Wine Quality (Red) dataset.**
>
> | $n/d$ | $\rho$ | Log loss MAE | Squared loss MAE | Betting loss MAE |
> | :--- | :--- | :--- | :--- | :--- |
> | **2** | 0.01 | 0.04096 $\pm$ 0.00010 | 0.04097 $\pm$ 0.00010 | **0.04073** $\pm$ 0.00009 |
> | | 0.02 | 0.05777 $\pm$ 0.00014 | 0.05776 $\pm$ 0.00014 | **0.05747** $\pm$ 0.00012 |
> | | 0.04 | 0.08152 $\pm$ 0.00016 | 0.08153 $\pm$ 0.00016 | **0.08134** $\pm$ 0.00016 |
> | **4** | 0.01 | 0.04071 $\pm$ 0.00010 | 0.04072 $\pm$ 0.00010 | **0.04047** $\pm$ 0.00009 |
> | | 0.02 | 0.05745 $\pm$ 0.00012 | 0.05744 $\pm$ 0.00012 | **0.05720** $\pm$ 0.00011 |
> | | 0.04 | 0.08121 $\pm$ 0.00015 | 0.08121 $\pm$ 0.00015 | **0.08104** $\pm$ 0.00015 |
> | **8** | 0.01 | 0.04041 $\pm$ 0.00008 | 0.04041 $\pm$ 0.00008 | **0.04021** $\pm$ 0.00008 |
> | | 0.02 | 0.05711 $\pm$ 0.00011 | 0.05711 $\pm$ 0.00011 | **0.05700** $\pm$ 0.00011 |
> | | 0.04 | 0.08092 $\pm$ 0.00014 | 0.08092 $\pm$ 0.00014 | **0.08083** $\pm$ 0.00014 |
>
> **Table 2: Comparison of average mean absolute error (MAE) on the Wine Quality (White) dataset.**
>
> | $n/d$ | $\rho$ | Log loss MAE | Squared loss MAE | Betting loss MAE |
> | :--- | :--- | :--- | :--- | :--- |
> | **2** | 0.01 | 0.04091 $\pm$ 0.00011 | 0.04091 $\pm$ 0.00011 | **0.04075** $\pm$ 0.00010 |
> | | 0.02 | 0.05763 $\pm$ 0.00013 | 0.05763 $\pm$ 0.00013 | **0.05743** $\pm$ 0.00012 |
> | | 0.04 | 0.08127 $\pm$ 0.00015 | 0.08127 $\pm$ 0.00015 | **0.08122** $\pm$ 0.00015 |
> | **4** | 0.01 | 0.04062 $\pm$ 0.00009 | 0.04062 $\pm$ 0.00009 | **0.04044** $\pm$ 0.00008 |
> | | 0.02 | 0.05731 $\pm$ 0.00012 | 0.05731 $\pm$ 0.00012 | **0.05713** $\pm$ 0.00011 |
> | | 0.04 | 0.08111 $\pm$ 0.00015 | 0.08112 $\pm$ 0.00015 | **0.08107** $\pm$ 0.00015 |
> | **8** | 0.01 | 0.04030 $\pm$ 0.00007 | 0.04029 $\pm$ 0.00007 | **0.04018** $\pm$ 0.00007 |
> | | 0.02 | 0.05701 $\pm$ 0.00011 | 0.05701 $\pm$ 0.00011 | **0.05691** $\pm$ 0.00010 |
> | | 0.04 | 0.08079 $\pm$ 0.00014 | 0.08080 $\pm$ 0.00014 | **0.08073** $\pm$ 0.00014 |
>
> **Reference:**
> Cortez, P., Cerdeira, A., Almeida, F., Matos, T., and Reis, J. Wine quality. UCI Machine Learning Repository, 2009.

---

### Official Review · Reviewer_ZQA9 · 2025-10-31

**Soundness:** 2
**Presentation:** 3
**Contribution:** 2
**Rating:** 2
**Confidence:** 4

**Summary:**

This paper gives a strong generalization bound for the basic task of $[0, 1]$-valued regression (or more generally settings where the target variable is a bounded real value). The main feature of these bounds is that they scale naturally with the intrinsic conditional variance (or noise level) $Var(y|x)$ of the target variable, so that the bound on the test loss of the ERM estimator grows tighter as the noise level decreases. Such a bound is described as a "variance-adaptive" or "second-order" generalization bound. This is similar in spirit to known "first-order" generalization bounds (aka "optimistic rates") that scale favorably with the optimal achievable loss (and variants thereof). Such bounds are stronger than naive uniform convergence bounds, which do not in general adapt to the optimal loss or variance at all.

The main contribution of the paper is a new method that achieves such a second-order guarantee for the $L^1$ loss by minimizing a surrogate "betting loss". The theoretical results are also supported by empirical experiments on synthetic data.

**Strengths:**

The problem studied by this paper, namely giving variance-adaptive / second-order generalization bounds, is a nice theoretical problem in statistical learning theory. The method proposed (minimizing a surrogate betting loss) seems appealingly simple and practical to use. The paper is generally written in a clear way.

**Weaknesses:**

The main weakness of the paper is that it does not seem to be aware of / engage with a large body of closely related prior work in generalization theory for supervised learning. This prior work is usually associated with keywords such as "optimistic rates", "optimal / localized generalization bounds", "benign overfitting", and "Moreau envelope theory", among others. The authors may claim that these works focus on so-called "first-order" bounds, whereas they are concerned with "second-order" bounds. But I would strongly contest this distinction in the setting considered in this paper, which is the _realizable_ setting (i.e. there exists $f^{\*}$ in the class such that $\\mathbb{E}[y|x] = f^{\*}(x)$). When you assume realizability, there is no longer an important distinction between "first-order" and "second-order" generalization bounds, because the expected conditional variance $\\mathbb{E}[\sigma_x^2]$ is precisely the best achievable square loss $\\min_{f \in F} \\mathbb{E}[(f(x) - y)^2] =: L_{sq}(f^{*})$. The latter is ostensibly a "first-order" quantity. Accordingly, it is not at all clear that the bounds in this paper are truly novel in light of existing theory.

It is true that the exact statement of the theorem differs in certain minor ways from typical statements of optimal generalization bounds (specifically, the LHS has $\\mathbb{E}[|f(x) - f^{*}(x)|]$ as opposed to the naive $L^1$ loss $\\mathbb{E}[|y - f(x)|]$, and the RHS has the optimal $L^2$ loss or variance). It is also possible that the exact distributional / model assumptions are a bit different. I am not necessarily claiming that the main theorem is a trivial corollary of existing bounds (I have not carefully checked this either way). But the main point is that the paper's contributions need to be carefully contextualized and compared against this existing work to be considered novel / significant.

Here are references to some particularly relevant related works:

Optimistic rates:
- Nathan Srebro, Karthik Sridharan, and Ambuj Tewari (2010). “Optimistic Rates for Learning with a Smooth Loss.”
  - The main theorem in this reference is very similar to the one considered in this paper, and may very well imply it with some additional work.
- Lijia Zhou, Frederic Koehler, Danica J. Sutherland, and Nathan Srebro (2021). “Optimistic Rates:
A Unifying Theory for Interpolation Learning and Regularization in Linear Regression.”
- Lijia Zhou, Frederic Koehler, Pragya Sur, Danica J. Sutherland, and Nathan Srebro (2022). “A
Non-Asymptotic Moreau Envelope Theory for High-Dimensional Generalized Linear Models.”
  - Please see the discussion of related work within this reference for a discussion of other related papers
  - [This Simons talk](https://www.youtube.com/watch?v=h1TGvwxRSd8) by Frederic Koehler discusses some of the results in this line of work

Older work on localized generalization bounds:
- Bartlett, Bousquet, Mendelson (2005). "Local Rademacher Complexities."
- Koltchinskii (2006). "Local Rademacher complexities and oracle inequalities in risk minimization."

**Questions:**

As discussed above, my recommendation to the authors would be to include a much more substantive discussion of and comparison with the prior work referenced above.

---

> ### Author Response · Authors · 2025-11-23
>
> We sincerely thank the reviewer for their detailed and insightful feedback.
>
> **[Relationship to prior work]**
>
> The reviewer rightfully highlighted the important connection to the literature on "optimistic rates" (Srebro et al., 2010; Zhou et al., 2021, 2022) and other works such as local rademacher complexity.
> We agree that these are relevant and should be discussed and will provide a detailed comparison in our revised manuscript.
> This revision would be minor because, as we explain below, these prior works do not diminish our paper's contribution.
>
> **1. The result in Srebro et al. (2010) implies a worse bound than ours.**
>
> The reviewer's key insight is that in our realizable setting ($f^\star \in \mathcal{F}$), the best achievable squared loss is precisely the expected conditional variance:
> $L^\star := \min_{f \in \mathcal{F}} \mathbb{E}[(f(x) - y)^2] = \mathbb{E}[(f^\star(x) - y)^2] = \mathbb{E}[\sigma_x^2]$. In conjunction with the fact that the excess risk under squared loss is exactly the $L^2$ generalization error: $L(\hat{f}) - L(f^\star) = \mathbb{E}[(\hat{f}-f^\star)^2]$, Srebro et al. (2010)'s main theorem, which bounds the excess risk for smooth losses, indeed implies a variance-adaptive $\tilde{O}(\sqrt{L^\star/n})$ bound on the $L^2$ generalization error:
> $$\mathbb{E}[(\hat{f}-f^\star)^2] \le \tilde{O}\left(\sqrt{\frac{\mathbb{E}[\sigma_x^2]}{n}}\right)$$
> However, our result is not a corollary of this, or its sharper, specialized counterparts, Zhou et al. (2021, 2022) (which achieve the same $\tilde{O}(n^{-1/2})$ rate order but with tighter constants for Gaussian linear regression). If one attempts to convert this $L^2$ bound to our paper's $L^1$ generalization error via Jensen's inequality:
> $$(\mathbb{E}[|\hat{f}-f^\star|])^2 \le \mathbb{E}[(\hat{f}-f^\star)^2] \le \tilde{O}(n^{-1/2})$$
> This yields a suboptimal rate of $\mathbb{E}[|\hat{f}-f^\star|] \le \tilde{O}(n^{-1/4})$.
>
>
> **2. Our Contribution is Fast Rates for $L^1$ generalization error.**
>
> Our work, by using a specialized betting loss, directly achieves a much faster $\tilde{O}(n^{-1/2})$ rate for the $L^1$ error (Theorem 3 and 5), which shows that our result is not a trivial corollary and that achieving a fast, variance-adaptive rate for the $L^1$ error is a distinct and more challenging problem requiring a novel approach.
>
> Our work is closely related to the literature on "optimistic rates" and "localized generalization bounds," which primarily focuses on bounding the excess risk (i.e., $L(\hat{f}) - L^\star$) and seeks to improve upon worst-case slow rates (typically $\tilde{O}(n^{-1/2})$) by exploiting low variance of the excess loss.
> Foundational work by Bartlett et al. (2005) and Koltchinskii (2006) established the machinery of Local Rademacher Complexities, showing that if the $L^2$ distance between any function $f$ and the optimal function $f^\star$ is upper bounded by the excess risk (a condition satisfied by squared loss), one can achieve faster rates. Srebro et al. (2010) proved that for smooth non-negative loss functions, the excess risk $L(\hat{f}) - L^\star$ scales with the optimal risk $L^\star$. Zhou et al. (2021) removed logarithmic factors for Gaussian linear regression, and Zhou et al. (2022) generalized these results using Moreau Envelope theory to handle model misspecification.
>
> One might ask if the squared loss could achieve a faster rate than Srebro's main theorem suggests.
> Section 3 of Srebro et al. (2010) summarizes the landscape of possible rates, showing that the $O(1/\sqrt{n})$ dependence is generally unavoidable for non-parametric or non-strongly convex settings. They demonstrate that a fast rate of $O(1/n)$ that is independent of $L^\star$ is possible only for smooth and strongly convex losses (like squared loss) in parametric settings.
>
> In this vein,
> Liang et al. (2015) introduced "Offset Rademacher Complexity" to study the excess risk under squared loss in agnostic settings.
> They confirmed (e.g., in their Lemma 10) that for parametric regression, the excess risk scales as $O(1/n)$, recovering the results of Rakhlin et al. (2015) without assuming boundedness of the noise or functions.
>
> However, squared loss minimization still has its fundamental limitations. Theorem 2 of Foster and Krishnamurthy (2021) proves lower bounds showing that the squared loss minimizer fundamentally fails to achieve the first-order fast $\tilde{O}(n^{-1/2})$ rate for $L^1$ metrics. Our work overcomes this limitation: by minimizing the proposed betting loss, we achieve the fast $\tilde{O}(n^{-1/2})$ variance-adaptive rate directly for the $L^1$ error.

---

> > ### Author Response · Authors · 2025-11-23
> >
> > **[Distinction between first-order and second-order]**
> >
> > We want to clarify why we used the phrase "second-order".
> > The reviewer correctly mentioned what people call as the first-order bound in supervised learning.
> > The reinforcement learning theory literature typically regresses onto the reward/loss values of the actions using a loss function.
> > Here, the loss and the loss function are two different things.
> > From the regression viewpoint, the loss is the label in supervised learning.
> > When an RL theory paper says first/second order, it is with respect to the loss (=label), not the loss function.
> > We follow this convention to say the first/second order bound in our work.
> > We will clarify this in the final version.
> >
> > **References**
> >
> > Bartlett, P. L., Bousquet, O., & Mendelson, S. (2005). Local Rademacher complexities.
> >
> > Foster, D. J., & Krishnamurthy, A. (2021). Efficient First-Order Contextual Bandits: Prediction, Allocation, and Triangular Discrimination.
> >
> > Koltchinskii, V. (2006). Local Rademacher complexities and oracle inequalities in risk minimization.
> >
> > Liang, T., Rakhlin, A., & Sridharan, K. (2015). Learning with Square Loss: Localization through Offset Rademacher Complexity.
> >
> > Rakhlin, A., Sridharan, K., & Tsybakov, A. B. (2015). Empirical entropy, minimax regret and minimax risk.
> >
> > Srebro, N., Sridharan, K., & Tewari, A. (2010). Optimistic Rates for Learning with a Smooth Loss.
> >
> > Zhou, L., Koehler, F., Sutherland, D. J., & Srebro, N. (2021). Optimistic Rates: A Unifying Theory for Interpolation Learning and Regularization in Linear Regression.
> >
> > Zhou, L., Koehler, F., Sur, P., Sutherland, D. J., & Srebro, N. (2022). A Non-Asymptotic Moreau Envelope Theory for High-Dimensional Generalized Linear Models.

---

### Official Review · Reviewer_yM3S · 2025-11-02

**Soundness:** 4
**Presentation:** 3
**Contribution:** 3
**Rating:** 8
**Confidence:** 2

**Summary:**

This paper studies regression problems where the target value is always between 0 and 1. Standard methods like squared loss aren't always the best fit for this specific case. While using log loss is an improvement and provides what's known as a "first-order bound"—meaning its performance guarantee is linked to the magnitude of the values being predicted—it still doesn't fully leverage the data's underlying characteristics, particularly when the amount of noise or uncertainty isn't uniform across all data points.
To address this, the authors propose a novel approach inspired by betting losses. The main result is that by minimizing this betting loss, their algorithm achieves a "second-order bound". Thus, the method's performance guarantee is tied to the actual variance in the data ($\sigma_x^2$), which can be much smaller than what first-order bounds depend on.
Interestingly, the algorithm is variance-adaptive; it achieves these tighter, more accurate results without needing any prior information about the noise levels in the data. The authors back up their theory with experiments, showing that their betting loss method consistently results in a lower mean absolute error (MAE) when compared to both traditional squared loss and the improved log loss, especially when the data has low variance.

**Strengths:**

- The authors proved improved bounds (in certain regimes) for a fundamental problem in learning theory.

- The results seem to be following in a non-trivial way.

**Weaknesses:**

- I found the intuition behind the betting loss function a bit unclear.

- The result requires that the label space $y$ is bounded in $[0,1]$.

**Questions:**

- Can you provide some more intuition behind the betting function?

- Can you say anything about unbounded losses? Relatedly, how does the bound scale with an upper bound on $y$ (different than 1)?

- One advantage that your result has is that it scales nicely with the noise of the label; for instance, assume that $y = f^*(x)$ w/ prob 1. Then, the error bound is $O(1/n)$ and increases smoothly as the noise in the label $y$ increases. Is my interpretation correct?

- Can you say how close to being optimal your bound is?

---

> ### Author Response · Authors · 2025-11-23
>
> We sincerely thank the reviewer for their positive assessment of our paper. We are glad they found that we "improved bounds... for a fundamental problem in learning theory". We would be happy to clarify the questions they raised.
>
> **1. Q: Can you provide some more intuition behind the betting function?**
>
> Think of our algorithm (Algorithm 1) as finding a hypothesis $f$ that is "unbeatable" by any other hypothesis $h$ from the same class $\mathcal{F}$. For any data point $x$, the bettor $h$ makes a "bet" ($h_x - f_x$), essentially wagering that the true label is in the direction of $h_x$. The bet is resolved against the "outcome" ($y - f_x$). The $\ln(1 + \ldots)$ term comes from the idea of log-wealth growth, where the bettor $h$ tries to maximize their wealth (the $max_h$ part) by finding a profitable betting strategy. Our algorithm finds the $f$ (the $min_f$ part) that no adversary can exploit. To be "unbeatable," $f$ must be extremely accurate, especially in high-variance regions. In these regions, ($y - f_x$) can be large, and a bettor $h$ could easily find a strategy to make large profits. This min-max pressure inherently forces $f$ to adapt to the local variance $\sigma_x^2$, which is the key to our second-order bound.
>
>
> **2. Q: Can you say anything about unbounded losses? Relatedly, how does the bound scale with an upper bound on $y$ (different than 1)?**
>
> This is indeed a very interesting future research direction! For unbounded losses, we believe some tail condition must be assumed.
>
> If the labels are in a different bounded interval, say $y' \in [0, B]$, our method can be applied via simple rescaling. We would define $y = y'/B$ and $f = f'/B$, which moves the problem back to the [0, 1] space.
>
> Let $f^{\star'}$ and $\sigma_x^{2'}$ be the true mean and variance in the original $[0, B]$ space. The rescaled problem has $f^\star = f^{\star'}/B$ and $\sigma_x^2 = \sigma_x^{2'}/B^2$.
> Our $L^1$ bound on the rescaled problem is $\mathbb{E}[|\hat{f} - f^\star|] \le \tilde{O}(\sqrt{\mathbb{E}[\sigma_x^2]/n} + 1/n)$. If we scale this bound back to the original $[0, B]$ space, the error becomes $\mathbb{E}[|\hat{f}' - f^{\star'}|] = B \cdot \mathbb{E}[|\hat{f} - f^\star|]$.
>
> Plugging everything in, the bound becomes:
> $$\mathbb{E}[|\hat{f}' - f^{\star'}|] \le B \cdot \tilde{O}\left(\sqrt{\frac{\mathbb{E}[\sigma_x^{2'}/B^2]}{n}} + \frac{1}{n}\right)$$
> That is,
> $$\mathbb{E}[|\hat{f}' - f^{\star'}|] \le \tilde{O}\left(\sqrt{\frac{\mathbb{E}[\sigma_x^{2'}]}{n}} + \frac{B}{n}\right)$$
>
> This shows that the leading variance-adaptive term scales correctly with the true variance $\mathbb{E}[\sigma_x^{2'}]$. The lower-order $1/n$ term scales linearly with $B$, which is reasonable as the magnitude of the problem's error scales with $B$.
>
> **3. Q: Scaling with noise**
>
> Yes, your interpretation is exactly correct.
>
> This is one of the key features of our proposed bound, and we are glad you found it to be an advantage.
>
> If $y = f^\star(x)$ with probability 1, this is the noiseless case. The conditional variance $\sigma_x^2 = 0$ with probability 1, so the expected variance $\mathbb{E}_x[\sigma_x^2] = 0$. As a result, the term with $\mathbb{E}_x[\sigma_x^2]$ in our bound vanishes, and we are left with the $O(1/n)$ rate you identified.
>
> As the noise increases, $\mathbb{E}_x[\sigma_x^2]$ grows from 0, and the first term $\tilde{O}(\sqrt{\mathbb{E}_x[\sigma_x^2]/n})$ "grows smoothly" to become the dominant part of the bound. This allows our bound to gracefully and adaptively interpolate between the fast $\tilde{O}(1/n)$ rate for noiseless problems and the standard $\tilde{O}(1/\sqrt{n})$ rate for worst-case, high-noise problems, all without needing to know the variance $\mathbb{E}_x[\sigma_x^2]$ in advance.
>
> **4. Q: Can you say how close to being optimal your bound is?**
>
> Our bound (w.r.t. $\tilde{O}(\sqrt{1/n})$) is indeed minimax-optimal in the adaptive sense.
> We can see this from our analysis of the linear class (Corollary 6).
> Our bound for the $d$-dimensional linear class is $\tilde{O}(\sqrt{\mathbb{E}[\sigma_x^2] \cdot (d/n)})$. The established minimax-optimal (non-variance-adaptive) rate for this class is $\tilde{O}(\sqrt{d/n})$.
> Our bound matches this minimax optimal rate (since $\mathbb{E}[\sigma_x^2] \le 1/4$). Noticeably, our bound is adaptive: it automatically becomes faster when the problem instance is "easy" (i.e., $\mathbb{E}[\sigma_x^2]$ is small) while never being worse than the known optimal rate. This is the best one can hope for from an adaptive bound.
>
> To our knowledge, the optimality of the rate with respect to the variance adaptivity is not known.

---

### Official Review · Reviewer_7NdS · 2025-11-05

**Soundness:** 3
**Presentation:** 3
**Contribution:** 2
**Rating:** 6
**Confidence:** 1

**Summary:**

The paper considers the problem of $[0,1]$-valued regression, where the goal is to learn a function $f^\star \in \mathcal{F} \subset \{f: \mathcal{X} \rightarrow [0, 1]\}$ that predicts $\mathbb{E}_x [y \mid x]$, under realizable setting. The authors argue that the standard squared loss is insensitive to variance, while the log loss (CE) can only achieve a first-order bound (scale with the variance proxy $f^*(x)(1 - f^*(x))$). The latter can be loose in some cases.

The above is also the motivation of this paper, which is trying to define a new loss for this problem that can actually achieve a second-order bound, scaling with the true conditional variance $\sigma_x^2 = \mathbb{E}_{y \mid x}[(y - f^*(x)^2]$. The authors give an affirmative answer by using the betting loss, which is well-known in the hypothesis testing framework.

**Strengths:**

The paper is clear and easy to follow.

**Weaknesses:**

Honestly, this paper gives me a hard time evaluating it fairly. On one hand, the __theoretical__ finding of this paper, though niche, is interesting enough. The proof technique is simple and natural, nothing to write home about. On the other hand, one might argue that the result only makes sense theoretically, and in practice, it is absolutely intractable due to the multi-level (not even bi-level) optimization nature. As a result, the experiments are very simple and only for demonstration purposes (the function class $\mathcal{F}$ is finite and contains only 21 hypotheses, the inner $\max_{\phi, c}$ cannot even be solved analytically, and has to use a grid-search). However, I would not consider it a weakness due to the theoretical nature of the results - the proposed loss is naturally intractable!


Given the above, I can only have lukewarm support for this paper with a low-confidence score. I leave the decision for the AC and other reviewers to evaluate the significance and appropriateness of this paper in a conference like ICLR. That being said, if I review this paper in AISTATS, I will definitely give it a better score with high confidence, and I truly believe that this draft should be submitted to AISTATS or ALT instead. But don't be disappointed! This draft is in good shape, and maybe other reviewers and AC will value this draft differently!

**Questions:**

No further questions. The problem setting is clear, and the result is well-presented. I only consider the experiments as add-ons, and in my view, even no experiment is acceptable for a paper with theoretical implications only. The proof technique is natural and sound - I checked it and found no issues.

---

> ### Author Response · Authors · 2025-11-23
>
> We thank the reviewer for their careful reading and for their positive feedback on the clarity of our paper and the "simple and natural" proof technique.
> The reviewer notes that our findings are "interesting enough" and "make sense theoretically".
> This is, in fact, our primary contribution.
>
> The main question we ask is: "Does a loss function exist that can achieve a second-order, variance-adaptive bound for [0,1]-valued regression, without prior knowledge of the variance?".
>
> Our paper's goal is to prove that the answer is "yes," and to show what such a loss function looks like. As stated in our introduction, our goal is to show that adapting to unknown variances can be a "free lunch, statistically speaking".
>
> We believe, as the reviewer seems to, that this theoretical finding is a valid contribution on its own. It identifies a new objective for variance-adaptive learning.
> The development of practical, scalable algorithms to optimize this objective is an exciting direction for future research, which we allude to in our conclusion.
>
> Finally, regarding the venue, we believe that ICLR is an appropriate fit for this work. The ICLR 2026 Call for Papers explicitly lists "learning theory" as one of the relevant topics. Our paper, which proposes a novel objective function to achieve fundamental improvements in generalization bounds, falls squarely within this scope.

---

> > ### Comment · Reviewer_7NdS · 2025-11-25
> > **Official Comment by Reviewer 7NdS**
> >
> > I thank the authors for the answers. My evaluation remains the same.

---

### Meta-Review · Area_Chair_uzVH · 2026-01-05

**Summary:**

This paper derives the second-order generalization bound, whose $O(\sqrt{1/n})$ term scales with the expected variance of the label noise, when the label is in $[0,1]$. This is an improvement over the existing first-order bound that scales with the expected magnitude of the label. To obtain the target bound, the authors introduce a new loss function called the betting loss and show that an empirical risk minimizer under the betting loss achieves the second-order generalization bound.

The most critical concern is about the missing related works, raised by Reviewer ZQA9. In the authors' response, they mentioned that the existing result (Sebro et al. 2010) also implies a second-order bound for the $L_2$ loss, and the main contribution of this paper is to derive a second-order bound for the $L_1$ loss. However, in the current manuscript, such a comparison is missing, and the main contribution can be misinterpreted as "the first second-order bound under the regression setup", not just for the $L_1$ loss. Due to this change in the main contribution, I think this paper requires major revision and should be reevaluated after clarifying the main contribution, adding more comprehensive discussions about existing works, and incorporating other unaddressed/partially addressed comments of reviewers.

**Reviewer Concerns:**

I think the following questions/concerns are not properly addressed:
- More discussions on the loss function (Reviewer YX2n),
- On computing the empirical risk minimizer in experiments (Reviewer YX2n),
- Intuition behind betting loss (Reviewer yM3S),
- Relationship to prior works (Reviewer ZQA9).

I couldn't find the answers to the first two questions. The intuition behind the betting loss has been briefly discussed, but it is unclear why such a loss can be used to derive the second-order bound. I recommend including additional discussions on the loss functions and proof sketch in the main body of the paper. The last comment has been answered, but it

**Reviewer Scores:**

I think Reviewer yM3S would maintain their score (reject), and the other reviewers may decrease their scores if they reevaluate the paper after reading the Reviewer ZQA9's review and the corresponding authors' response.

---

### Decision · Program_Chairs · 2026-01-26

Reject